# The Effects of Wind Farm Wakes on Freezing Sea Spray in the Mid-Atlantic Offshore Wind Energy Areas

David Rosencrans[1,2], Julie K. Lundquist[1,2,3], Mike Optis[2,4], and Nicola Bodini[2]

[1]Department of Atmospheric and Oceanic Sciences, University of Colorado, Boulder, 80303, USA
[2]National Renewable Energy Laboratory, Golden, 80401, USA
[3]Renewable and Sustainable Energy Institute, Boulder, 80303, USA
[4]Veer Renewables, Courtenay, V9N 9B4, Canada

*Correspondence to*: David Rosencrans (David.Rosencrans@Colorado.edu)

**Abstract**

The U.S. is expanding its wind energy fleet offshore where winds tend to be strong and consistent. In the mid-Atlantic, strong winds, which promote convective heat transfer and wind-generated sea spray, paired with cold temperatures can cause ice on equipment when plentiful moisture is available. Near-surface icing is induced by a moisture flux from sea spray, which poses a risk to vessels and crews. Ice accretion on turbine rotors and blades occurs from precipitation and in-cloud icing at temperatures below freezing. Ice accretion induces load and fatigue on mechanical parts which reduces blade performance and power production. Thus, it is crucial to understand the icing hazard across the mid-Atlantic. We analyze Weather Research and Forecasting model numerical weather prediction simulations at coarse temporal resolution over a 21-year period to assess freezing events over the long-term record and at finer granularity over the 2019–2020 winter season to identify the post-construction turbine impacts. Over the 2019–2020 winter season, results suggest that sea-spray–induced icing can occur up to 67 hours per month at 10 m at higher latitudes. Icing events during this season typically occur during cold air outbreaks (CAO), which are the introduction of cold continental air over the warmer maritime surface. During the 2019-2020 winter season, CAO lasted a total duration of 202 hours. While not all FSS events occurred during CAO over the 21 year period, all CAO events had FSS present. Further, we assess the turbine–atmosphere impacts of wind plant installation on icing using the fine-scale simulation data set. Wakes from large wind plants reduce the wind speed which mitigates the initiation of sea spray off white-capped waves. Conversely, the near-surface turbine-induced introduction of cold air in frequent wintertime unstable conditions enhances the risk for freezing. Overall, the turbine–atmosphere interaction causes a small reduction of FSS hours within the wind plant areas, with a reduction up to 15 hours in January at the 10 and 20 m heights.

## 1 Introduction

The offshore wind energy industry is undergoing rapid growth to supply emissions-free energy to the electrical grid. In the U.S., offshore capacity targets are approaching 40 GW by 2040 (Musial et al., 2022). Capacity expansion into relatively cold offshore regions will subject turbines to harsher wintertime conditions, which necessitate an understanding of the hazards that marine icing poses to offshore wind turbines, service vessels, and crew safety.

38

Ice accretion reduces the aerodynamic efficiency of the turbine blade, which hinders energy capture and annual energy production (Battisti et al., 2006; Kraj and Bibeau, 2010; Wei et al., 2020). Ice can remain on the rotors even after freezing conditions end, as slow natural processes such as ice shedding and melting extend the limitation to energy yield (Gao and Hong, 2021). One study found that excessive icing induced a power loss of 63 % for a single turbine over a 51-h icing event (Gao and Hu, 2021). Faster winds during cold front passages can enhance wind-energy supply during high-load cold-weather events, although, following frontal passages, the combination of cold temperatures and slow wind speeds may pose severe challenges for utility grid planners (Novacheck et al., 2021). Despite the energy losses from ice accretion, various strategies can mitigate or even prevent ice accretion altogether (IEA, 2018; Madi et al., 2019). While turbine blade icing is well studied (IEA, 2018; Martini et al., 2021; Contreras Montoya et al., 2022), icing near the turbine base, affecting operations and maintenance activities, is not.

The leading causes for low-level offshore icing are wave-impact and wind-induced sea spray (Dehghani-Sanij et al., 2017). Sea spray provides nuclei for ice clouds at high latitudes where airborne dust is sparse, being lofted by bursting bubbles and droplets from white-capped waves (Russell, 2015; Dehghani-Sanij et al., 2017). Ice accumulation from spray raises the center of gravity of ships, which can cause loss of stability and lead to capsizing (Guest and Luke, 2005). Observations suggest that the liquid droplets torn off of white caps, referred to as spume, experience a marked increase in concentration with strong winds above 9 m s$^{-1}$ (Ross and Cardone, 1974; Monahan et al., 1983; Monahan and MacNiocaill, 1986). Further, spray particles more easily supercool with cold sea surface temperatures (SST) below 7° C and at air temperatures below the freezing point for saline ocean water at −1.7° C (U.S. Navy, 1988; Guest and Luke, 2005). Ice accumulation is believed to have caused the recent losses of three ships, including 1) the *Destination*, which sank near St. George Island, Alaska in 2017 (Kraegel, 2018); 2) the *Scandies Rose*, which sank southeast of Kodiak, Alaska, in 2019 (NTSB, 2021); and 3) the *Onega*, which sank in the Barents Sea in 2020 (Nilsen, 2020). To mitigate ice-induced accidents, inclement weather forecasts are furnished for coastal waters. A Coastal Waters Forecast, delivered by the National Weather Service, will contain a "freezing spray advisory" if freezing water droplets can accumulate on vessels due to a combination of SST, wind speed, air temperature, and vessel motion (Glossary - NOAA's National Weather Service, 2023). At accumulation rates greater than 2 cm h$^{-1}$, the advisory becomes a "heavy freezing spray watch".

Wind turbines can modify the amount and severity of icing conditions via competing effects. Enhanced turbulence caused by spinning blades transports heat from aloft to lower altitudes within the rotor-swept region or near the surface. In stable stratification, warmer potential temperatures are transported downward, which introduces a near-surface warming effect, and vice versa in unstable conditions (Fitch et al., 2013; Rajewski et al., 2013; Xia et al., 2016; Siedersleben et al., 2018; Tomaszewski and Lundquist, 2020). However, recent research suggests taller turbines may reverse this phenomenon (Golbazi et al., 2022) depending on the depth of the atmospheric boundary layer (Quint et al., 2024). As the winter months feature more frequent unstable stratification along the U.S. East Coast (Bodini et al., 2019), turbine-induced cooling may increase the potential for near-surface freezing. In contrast,

turbines harness momentum from the flow, which reduces the downwind wind speed (Nygaard, 2014; Platis et al.,
2018; Schneemann et al., 2020). A reduction in wind speed conversely reduces the potential for icing (Dehghani-
Sanij et al., 2017). Thus, it is crucial to understand how large-scale wind deployment across the mid-Atlantic will
modify the regularity and intensity of freezing sea spray (FSS) conditions.

Herein, we employ numerical weather prediction modeling to quantify the baseline offshore icing risk and the
wind plant post-construction effects. Section 2 outlines the modeling setup and discusses the techniques for
discerning icing conditions and cold air outbreak events. Section 3 reports results for the spatiotemporal icing risk,
causal factors, and the adjustments by wind plants. Section 4 offers concluding remarks and discussion.

**2 Methods**
**2.1 NOW-23**
We explore annual variability of FSS conditions using the 2023 National Offshore Wind (NOW-23) data set
(NREL, 2020; Bodini et al., 2024). This data set quantifies wind resources spanning all offshore regions of the
United States for more than 20 years using the Weather Research and Forecasting (WRF) model version 4.2.1
(Powers et al., 2017). We acquire model output at an hourly temporal resolution for the 21-year period from 01
January 2000 to 31 December 2020. A parent domain feeds into an inner nested domain with horizontal grid
resolutions of 6 km and 2 km, respectively. Both domains incorporate a vertical grid resolution of 5 m near the
surface with stretching to 45 m aloft, using 61 vertical levels up to a 50 hPa top. The European Centre for Medium
Range Weather Forecasts 5 Reanalysis (ERA5) dataset supplies hourly initial and boundary conditions at a 30 km
resolution to WRF (Hersbach et al., 2020). NOW-23 employs the MYNN2 planetary boundary layer and surface
layer (Nakanishi and Niino, 2006) schemes, eta microphysics (Ferrier et al., 2002), the Noah Land Surface Model
(Tewari et al., 2004), the rapid radiative transfer model for shortwave and longwave radiation (Iacono et al., 2008),
and the Kain–Fritsch cumulus parameterization (Kain, 2004) in the outmost domain only. For the mid-Atlantic
region, NOW-23 was validated against observations from three ZephIR ZX300M floating lidars (Pronk et al., 2022).
**2.2 NOW-WAKES**
We explore the seasonal variability and impacts of wind plants on icing conditions using high-fidelity numerical
weather prediction simulations over the period 01 September 2019 to 31 August 2020. These validated WRF version
4.2.1 simulations are described in detail in Rosencrans et al., (2024) but are summarized here for the reader's
convenience. This period is chosen for the availability of lidar measurements for validation of the wind speed
profile. A parent domain hosts an inner nest with horizontal grid resolutions of 6 km and 2 km, respectively (Figure
1). Both domains include a vertical grid resolution of 10 m near the surface with stretching aloft, using 54 vertical
levels up to a 50 hPa top. The inner domain outputs data at an instantaneous history file frequency of 10 minutes.
Constant time steps are set to 18 s and 6 s in the outer and inner domains, respectively. Initial and boundary
conditions are also supplied by the hourly 30 km ERA5 dataset (Hersbach et al., 2020). Lower boundary conditions
are provided as SST by the UK Met Office Operational Sea Surface Temperature and Sea Ice Analysis dataset
(Donlon et al., 2012) and show good agreement during validation against mid-Atlantic bight buoys (Redfern et al.,
2023). Physics parameterizations include the MYNN2 planetary boundary layer and surface layer (Nakanishi and
Niino, 2006), the Noah Land Surface Model (Niu et al., 2011), the New Thompson microphysics (Thompson et al.,
2008), the rapid radiative transfer model for longwave and shortwave radiative transfer (Iacono et al., 2008), and the
Kain–Fritsch Cumulus (Kain, 2004) schemes. The Kain–Fritsch cumulus parameterization applies to the parent
domain only. We incorporate spectral nudging to relax model output toward the ERA5 boundary conditions in the
inner domain. We apply a cutoff wavenumber of 3 (Gómez and Miguez-Macho, 2017), above which model
dynamics may resolve freely. No nudging is applied beneath the boundary layer height.

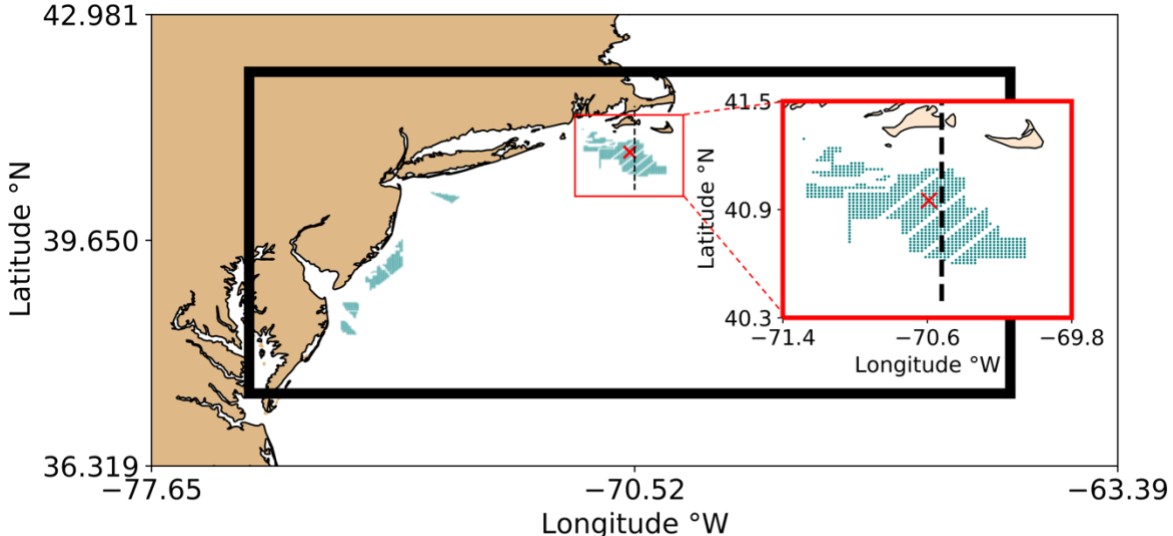


**Figure 1. Modeling domains. The entirety of the outer domain with inner domain is shown, outlined by the black
rectangle. The red square is zoomed in on the Rhode Island–Massachusetts (RIMA) block to enhance visibility. Turbines
are shown as teal dots. The red "X" indicates the point of interest (POI) where time series are acquired. The dashed black
line is a cross section extending through the RIMA block.**


We incorporate the effects of wind turbines using the WRF wind farm parameterization (WFP) (Fitch et al.,
2012). WFP simulations feature wind plant layouts of the lease areas and include 1,418 turbines (Figure 1, Table 1).
The WFP incorporates the effects of turbines by implementing a drag-induced deceleration of wind flow and an
addition of turbulence at model levels intersecting the rotor area. We execute WFP simulations adding both 0 % and
100 % turbulent kinetic energy (TKE) (Rosencrans et al., 2024), although a smaller value of 25 % in some cases
agrees better with neutrally stratified large-eddy simulations (Archer et al., 2020). Differences in the number of icing
hours between 0 % and 100 % added TKE are slight, so we report those from 100 % added TKE only. Thus, for the
remainder of this article we refer to the 100 % added TKE simulation as "WFP". This work utilizes 12 MW GE
Haliade wind turbines with a 138 m hub height and 215 m rotor diameter, which are scaled by Beiter et al. (2020)
from a 15 MW reference turbine. We carry out separate simulations using both no wind farms (NWF) and wind
farms (WFP) for the full year-long period from 01 September 2019 to 31 August 2020 (Table 1).

**Table 1. List of NOW-WAKES WRF simulations characterized by turbine characteristics. The simulation period spans 01 September 2019 to 01 September 2020.**

| Simulation type | Acronym | Turbine rated power | Added TKE | # Turbines |
|---|---|---|---|---|
| No Wind Farms | NWF | N/A | N/A | 0 |
| Wind Farm Parameterization | WFP | 12 MW | 100 % | 1,418 |

### 2.3 Icing hours detection

Ice accretion occurs when supercooled water freezes upon contact with objects. The largest contributions to sea spray icing are provided by the bursting of bubbles and advection of spray from white-capped waves (Dehghani-Sanij et al., 2017). In the presence of moisture, three key variables dictate offshore freezing conditions: wind speed, SST, and air temperature (Overland et al., 1986; Overland, 1990; Guest and Luke, 2005; Dehghani-Sanij et al., 2017; Line et al., 2022).

We detect FSS conditions following common thresholds defined by the latter studies (Guest and Luke, 2005; Dehghani-Sanij et al., 2017; Line et al., 2022). These criteria require 1) wind speeds in excess of 9 m s$^{-1}$, 2) air temperatures below −1.7° C, and 3) SST less than 7° C. Air temperature and SST thresholds can range between −2° C and −1.7° C and between 5° C to 8.9° C, respectively, as reviewed by Dehghani-Sanij et al., (2017). As such, we provide a sensitivity assessment for the full range (Appendix B). The surface skin temperature (WRF output variable "TSK") is assessed because the SST field inherits coarse blocks of missing data around coastlines from the ERA5 dataset. The resulting spatial maps are masked by the land use (WRF output variable "LU_INDEX") to ensure that icing conditions over land are not counted. The number of 10 min timestamps where these criteria are met each month are recorded for all simulations. As sea spray often lofts to between 5 and 20 m above sea level (Dehghani-Sanij et al., 2017), we quantify sea spray–induced icing at the 10 and at 20 m heights. For the 20 m conditions, we use 20 m air temperatures but use 10 m wind speeds as those winds have been linked to the generation of spray off white-capped waves (Dehghani-Sanij et al., 2017; Guest and Luke, 2005; Line et al., 2022; Ross and Cardone, 1974; Monahan et al., 1983; Monahan and MacNiocaill, 1986).

Due to the height constraint of sea spray particles, we consider both precipitation-based and in-cloud icing at the 138 m hub height by assessing different criteria for 1) the nonzero presence of liquid rain water (WRF variable "QRAIN") that may become supercooled at temperatures less than 0° C, 2) ice (WRF variable "QICE"), and 3) the aggregation from snow (WRF variable "QSNOW") (Parent and Ilinca, 2011; ISO, 2017). Further, we detect cloud or fog formation when 4) the relative humidity (RH) is greater than or equal to 100% following:

$$e_s = e_0 \exp\left[\frac{b(T - T_1)}{(T - T_2)}\right] \quad (1)$$

$$w_s = \frac{\epsilon e_s}{p - e_s} \quad (2)$$

$$RH = \frac{w}{w_s} \times 100\% \tag{3}$$

where $e_s$ is the saturation mixing ratio, $e_0$ is 6.112 mb, $b$ is 17.67, $T_1$ is 273.15 K, $T_2$ is 29.65 K, $T$ is the air temperature, $\epsilon$ is 0.622, $p$ is the atmospheric pressure, and $w$ is the mixing ratio (WRF output "QVAPOR") (Stull B., 1988). None of the aforementioned criteria must occur at the same time for icing to occur. However, we require that one must occur in conjunction with an air temperature less than 0° C for an icing event.

### 2.4 Ice accumulation rate

A predictability function assesses the likelihood for freezing in the presence of sea spray. We assess the predictability of icing conditions at the point of interest (POI) in the Rhode Island/Massachusetts (RIMA) block (Figure 1) separately from the NOW-WAKES and the NOW-23 datasets. The predictability ($PR$) for sea spray–induced ice formation follows:

$$PR = \frac{V_a(T_f - T_a)}{1 + 0.4(T_s - T_f)} \tag{4}$$

where $V_a$ is the wind speed, $T_f$ is the temperature threshold of $-1.7°$ C, $T_a$ is the air temperature, and $T_s$ is the SST (Guest and Luke, 2005; Overland et al., 1986; Overland, 1990). A humidity variable is not present in Eq. (4) due to the assumption that sea spray introduces a constant source of moisture during fast winds. A group of successive timestamps with nonzero PR are considered the same event. Separate flagged timestamps occurring within 24 hours of each other span the same synoptic regime (Winters et al., 2019), and so the entire duration between the two flagged timestamps is considered one event. We additionally tested a threshold of 72 hours to account for synoptic conditions spanning a longer duration but found that one FSS event lasted for over a week and our three FSS criteria were only met 8 % of the time during the event. As such, the 72-h threshold was not justified.

**Table 2. Icing rate by PR. Rows delineate the PR value, icing class, and ice accretion rate. Columns delineate the icing rate per PR range. From Guest and Luke (2005).**

| PR | <0 | 0–22.4 | 22.4–53.3 | 53.3–83.0 | >83.0 |
|---|---|---|---|---|---|
| Icing Class | None | Light | Moderate | Heavy | Extreme |
| Icing Rate [cm h$^{-1}$] | 0 | <0.7 | 0.7–2.0 | 2.0–4.0 | >4.0 |

The magnitude of PR can determine the rate of ice accretion (Table 2). The ice accretion rates are a general guideline developed for 20 to 75 m long vessels; specific rates depend on the type of ship, its load, heading relative to the prevailing wind direction, and its handling characteristics (U.S. Navy, 1988; Guest and Luke, 2005). For instance, a larger ship requires faster winds and taller waves for sea-spray–induced ice to accumulate on a higher deck but is more vulnerable to the prevailing wind direction due to reduced maneuverability. It is not known how these icing rates would apply to wind turbines or to the vehicles used to access offshore wind turbines.

**2.5 Cold air outbreak detection**

Freezing conditions can be stimulated by the advection of cold continental air over a warmer maritime surface. The resulting temperature profile causes thermal instability, which can induce filamentary convective rolls that align to make cloud "streets" with parallel columns of ascending and descending air that transform into open convective cells further offshore (Geerts et al., 2022). Convective rolls can be used to identify cold air outbreak (CAO) (Atkinson and Wu Zhang, 1996; Geerts et al., 2022) and may also contribute moisture for in-cloud icing if the lifting condensation level is at or below rotor-swept heights. A quantitative approach proposed by Vavrus et al. (2006) identifies a cold air outbreak (CAO) by the magnitude and duration of anomalous air temperature, which we apply at the POI (Figure 1). This strategy requires that the near-surface temperature be at least 2 standard deviations below the wintertime average following Eq. (5):

$$T < \bar{T} - 2(\sigma) \tag{5}$$

where $T$ is the 2 m temperature, $\bar{T}$ is the average 10 m temperature over the entire wintertime period, and $\sigma$ is the standard deviation. The wintertime period spans November through March at a 10 min frequency to account for all non-zero-freezing predictability events. Again, successive timestamps with detected CAO are considered a single event, and separate events occurring within a 24 h span are conglomerated into the same event.

**2.6 Atmospheric stability**

Turbulence from wind turbines modifies the near-surface temperature based on the atmospheric stability or stratification. We calculate the modeled atmospheric stability using the Obukhov Length ($L$) (Monin and Obukhov, 1954) (Eq. 6), which delineates the height above the surface at which buoyant turbulence equals mechanical shear production of turbulence, at a point centered on the RIMA block of lease areas:

$$L = -\frac{u_*^3 \overline{\theta_v}}{\kappa g \left( \overline{w'\theta_v'} \right)} \tag{6}$$

where $u_*$ (UST in WRF output) is the friction velocity, $\theta_v$ is the virtual potential temperature, $\kappa$ is the von Kármán constant of 0.4, g is gravitational acceleration of 9.81 m s$^{-1}$, and $\overline{w'\theta_v'}$ (HFX in WRF output) is the surface dynamic heat flux converted into kinematic heat flux. Negative lengths between 0 m and −500 m imply unstable stratification due to a positive heat flux (Gryning et al., 2007; Archer et al., 2016). Conversely, lengths between 0 m and 500 m imply stable stratification due to a negative heat flux. Lengths approaching negative or positive infinity imply neutral stratification, as buoyancy is no longer a dominating factor. Each 10 min timestamp from the NWF run is assigned a stability classification from November 2019 to March 2020.

**3 Results**

**3.1 Spatial variability of icing conditions**

The prevalence of icing conditions exhibits regional variability. The commonality of icing increases toward higher latitudes and near the coast where cold continental air advects over the ocean during the winter (Figure 2). In general, the spatial icing pattern during the 2019–2020 winter season (Figure 2a) matches well with the pattern over the 21-year period (Figure 2b) although the 2019–2020 season is relatively mild compared to other winters (Figure 2, Figure 3a). Icing conditions shadow the mid-Atlantic coast but occur less often along the New Jersey Bight where

wind speeds decrease and air and sea temperatures warm. The prevalence of freezing conditions extends furthest
offshore southeast of Nantucket and enhances in the Long Island Sound; both regions feature local minima in mean
January 2020 SST less than 5° C. The Long Island Sound is flanked by land to the north and south which amplifies
the presence of cold air. In addition, mean wind speeds maximize to the east of Cape Cod and Nantucket (Bodini et
al., 2024) which increases the number of hours that wind-generated spray is present. Finally, the cyclonic current in
the Gulf of Maine transports water southward. East of Cape Cod, this current bifurcates around the Georges Bank,
and a branch feeds cold water into the mid-Atlantic (Chapman et al., 1986). The number of icing hours may be
further exacerbated when predominant northerly winter winds instigate onshore Ekman transport toward the coast,
which is favorable for downwelling (Shcherbina and Gawarkiewicz, 2008b). However, downwelling is not always
supported, as the mixed layer stratification is dominated by salinity (Shcherbina and Gawarkiewicz, 2008a), leaving
a cold pool near the surface.

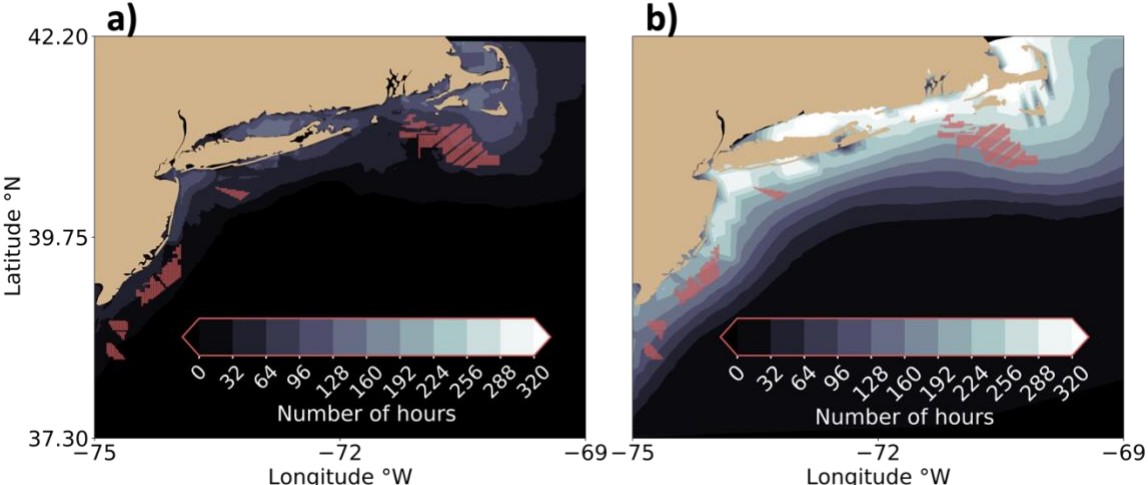


**Figure 2. The number of hours FSS conditions occur at 10 m during (a) the November 2019 to March 2020 period in**
**NWF and (b) the mean November to March period from 2000 to 2020 in NOW-23. Lighter contouring indicates more**
**freezing hours. Red dots represent turbine locations but do not exist in (a) or (b) and are shown for reference.**

Icing conditions exhibit seasonal variability in NWF, starting at 0 hours in November, increasing through the
winter, and falling to 0 again by April at all heights (Figure 3 and Figure. A1–A3). At the 10 m altitude, FSS
conditions occur most often in January, up to 67 hours, with an offshore spatial extent of 59,292 km$^2$, or 12.3 times
the area of the wind plants. At 20 m, FSS conditions also occur most often in January, up to 68 hours, covering a
total area of 61,736 km$^2$, or roughly 12.8 times the area of the wind plants (Figure A2). The 138 m hub height attains
the largest maximum of 119 hours during January in the Gulf of Maine and to the east Cape Cod (Figure A3), with
an offshore spatial extent of 291,012 km$^2$, or 60.2 times the area of the wind plants.

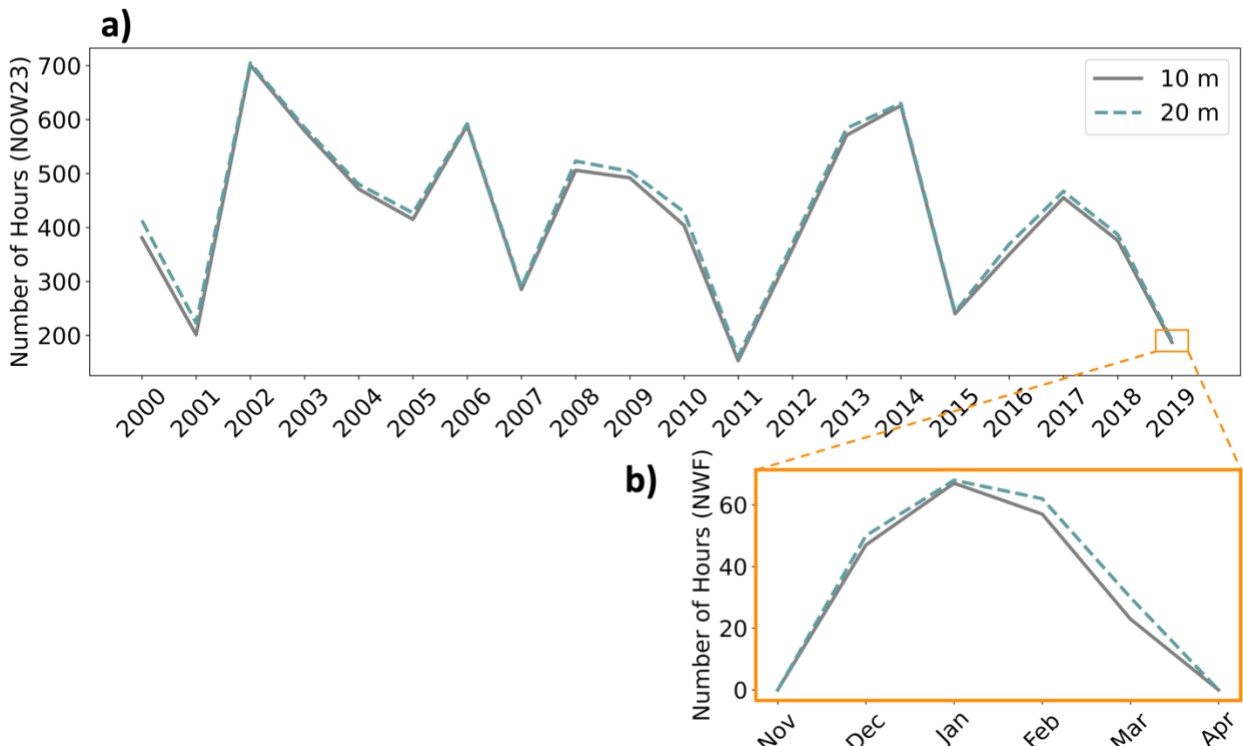

**Figure 3. The maximum number of FSS hours over the OCS (a) annually and (b) seasonally in NOW-23. The zoomed orange cutout shows the seasonal variation over the 2019–2020 winter.**

The 2019–2020 winter season was one of the mildest compared to other winters (Figure 3a), as assessed using the FSS detection criteria (Section 2.3). This winter season had few icing hours compared to other winters over the 21-year period, reaching 194 hours in NWF or 187 hours in NOW-23 at 10 m. At 20 m, the 2019–2020 winter season contains 210 hours in NWF or 191 hours in NOW-23. The greatest number of icing hours occurs during the 2002–2003 season, with 701 total hours at 10 m and 705 hours at 20 m. While the 21-year slope shows a decrease, it is not statistically significant using the Mann–Kendall (M–K) test (Hussain and Mahmud, 2019). P-values for the maximum number of icing hours (found across the OCS) (Figure 3a) and for the number of hours at the POI (Figure 1) are 0.20 and 0.12, respectively. We additionally applied the seasonal M–K test (Hirsch et al., 1982) to account for upward and downward trends throughout the year on monthly mean PR, monthly maximum PR, and the monthly total number of icing hours at the POI. Neither test returned a statistically significant trend.

### 3.2 Icing conditions and cold air outbreak

Investigating all events with a non-zero PR at the POI (Figure 1) reveals similar synoptic trends. We identify seven events with FSS conditions with a total duration of 253 hours from November 2019 to March 2020. All times during the 2019–2020 winter period with nonzero PR contain light ice accumulation of less than 0.7 cm h$^{-1}$ (Table 2). During each FSS event, higher relative pressure resided to the southwest throughout the Great Plains, Appalachia, or the Great Lakes with lower relative pressure to the northeast around Novia Scotia and

Newfoundland. In the Northern Hemisphere, winds flow with higher pressure to the right and lower pressure to the
left (Wallace and Hobbs, 2006). This flow regime results from the balance between the pressure gradient force and
the Coriolis force, which is a force introduced into the equations of motion to account for acceleration on a non-
inertial rotating reference frame (Ferrel, 1856). The largest pressure gradient forces occurred during the two January
events reaching 4 hPa per 100 km, or roughly 4 times the pressure gradient force required for a 10 m s$^{-1}$ geostrophic
wind in the midlatitudes. Most events feature a cold front in the mid-Atlantic. This pressure regime directs quasi-
geostrophic flow near the surface toward the southeast, introducing cold continental air offshore. During the winter,
the prevailing wind direction is northwesterly across the mid-Atlantic OCS (Bodini et al., 2019) because regions of
land mass feature higher surface pressure than the surrounding ocean and the Bermuda High retreats to the east.

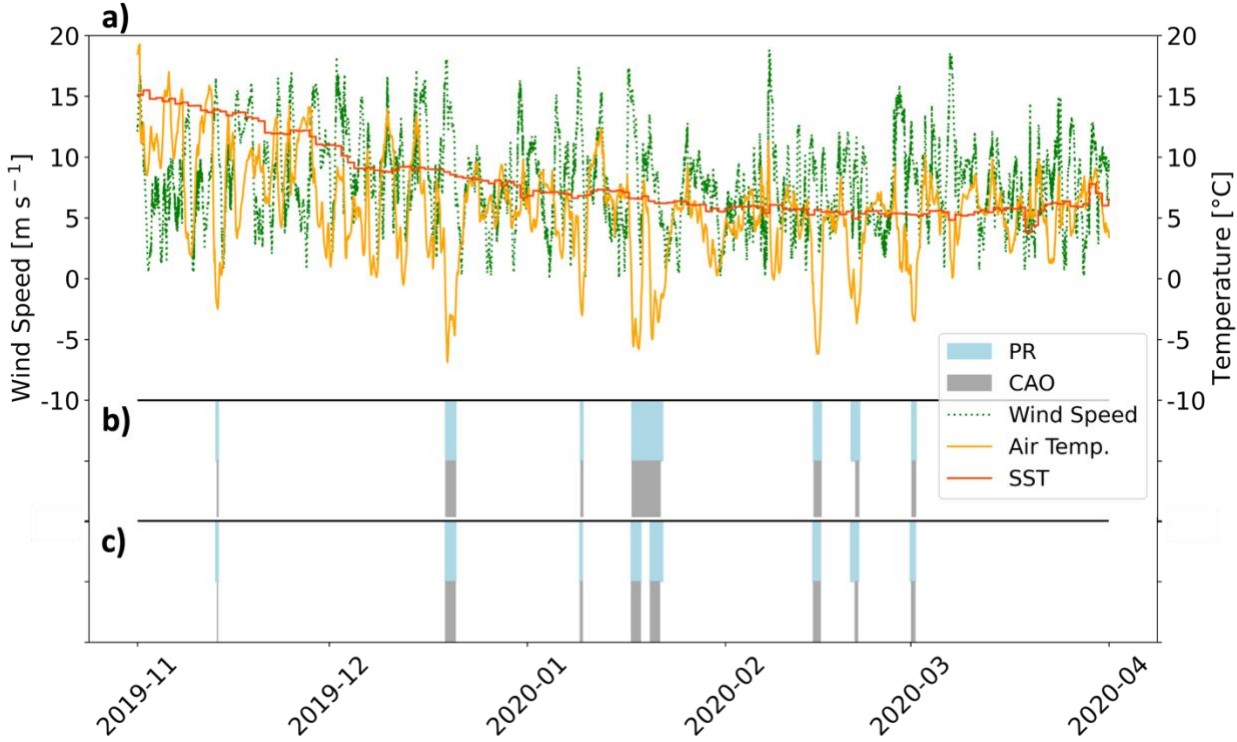

**Figure 4. (a) Time series of wind speed (green dotted), 10 m air temperature (orange), and SST (red) from November**
**2019 to April 2020 at the downwind edge of Vineyard Wind (Figure 1). Light-blue shading indicates the duration of**
**nonzero PR, and gray shading indicates the duration of detected CAO from (b) NWF and (c) NOW-23.**
All FSS events, assessed using PR, coincide with CAO. We detect seven CAO events in NWF with a total
duration of 202 hours (Figure 4b). The mean duration of CAO events (29 hours) are seven hours shorter than FSS
events (36 hours), with 80 % of flagged FSS timestamps having CAO present.
Common between events are fast wind speeds and cold 10 m air temperatures; SST plays a secondary role for
its weak temporal variability (Figure 4a). The average wind speed during FSS events is 10 m s$^{-1}$ with gusts
exceeding 15 m s$^{-1}$ during four events. Nonzero PR does not occur until after the wind speed peaks, when cold air
temperatures sweep in, averaging minimum temperatures of −4.5° C (Figure 4a). This wind speed–temperature
dynamic can pose a challenge for grid planners if wind energy generation reduces during periods of high demand for
residential and commercial heating, especially in a future scenario with electrification of space heating.

During the 2019–2020 winter in the NOW-23 dataset, eight total events are flagged as candidates for FSS
because the longest event in January 2020 (Figure 4b) is split among two separate events; all eight events have a
corresponding CAO (Figure 4c). Over the 21-year period, *all* CAO events occur in conjunction with an FSS event
(positive PR) (Fig. C1–Fig. C20). However, many FSS events occur without CAO present meaning that CAO is
only one of the drivers, and large interannual variability can exist. For instance, while 100 % of CAO timestamps
concur with FSS during the 2011–2012 season, only 10 % do during the 2013–2014 season.

The 2019–2020 winter ice accumulation rate is similar to other winters. The average PR during freezing events
from 2019 to 2020 is 4.3, which corresponds to a light ice accumulation rate of less than 0.7 cm h$^{-1}$ (Table 2). Over
the 21-year period, the average PR among events is 8.1, which corresponds to the same accumulation rate. The
2003–2004 winter period features the greatest mean PR of 15.7, which also corresponds to a light ice accumulation
rate. During this winter, a moderate risk for icing occurred 18 % of the time, and a heavy risk occurred 3 % of the
time, corresponding with icing rates between 0.7–2.0 cm h$^{-1}$ and 2.0–4.0 cm h$^{-1}$, respectively, and possibly
triggering heavy freezing spray watches in the NWS advisory.
Synoptic-scale teleconnection patterns can impact the likelihood of icing conditions. From December 2003 to
March 2004, the Pacific North Atlantic (PNA) cycle was positive. During the positive phase of PNA, a relative high-
pressure anomaly with anticyclonic wind flow exists over the western US that is conducive to northwesterly
transport of cold air over the East Coast (Vavrus et al., 2006).  In addition, the entire November 2003 to March 2004
period featured a positive El Niño-Southern Oscillation (ENSO) index. Positive ENSO has been attributed to cooler
SSTs across the mid-Atlantic and northeasterly winds which advect cold air from the north (Alexander and Scott,
2002). Other teleconnection patterns, including the Arctic Oscillation and North-Atlantic Oscillation switched signs
during this winter and are not discussed in greater detail.

**3.3 Modifications by wind plants**
The near-surface cooling effect by rotor turbulence provides a subtle effect on freezing conditions. In unstable
conditions, which occur 64 % of the time from November 2019 through March 2020 in NWF assessed at the POI,
wind turbines introduce near-surface cooling, which could increase the likelihood of freezing. Mean cooling and
warming during unstable conditions reach magnitudes up to −0.041 K at the surface and 0.022 K within the rotor-
swept region, respectively, along a cross section extending through the RIMA block (Figure 1, Figure 5b). During
stable conditions, which occur 25 % of the time from November through March, cooling aloft reaches up to −0.34
K, and near-surface warming reaches 0.26 K (Figure 5a). Near-surface cooling exists adjacent to the wind plant
cluster (Xia et al., 2016).

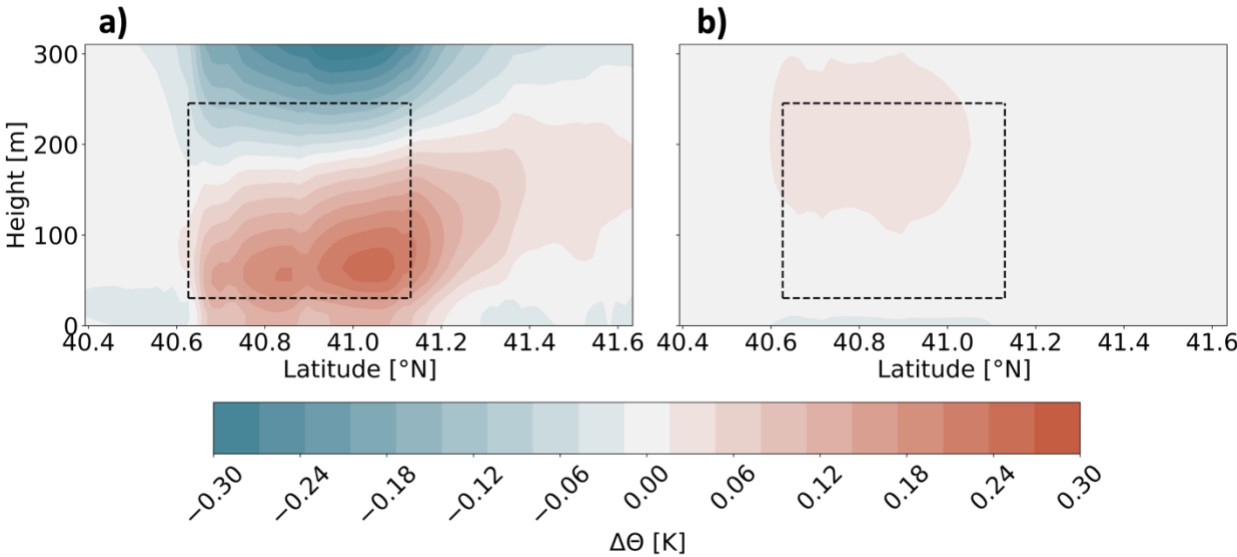

**Figure 5. The mean (WFP-NWF) potential temperature difference during (a) stable stratification and (b) unstable stratification, from November 2019 to March 2020. The cross section spans the RIMA block of lease areas (Figure 1). Red contouring indicates warming, and blue indicates cooling. Dashed lines outline the wind plant area and rotor-swept region.**

The reduction of wind speeds in the wake modifies the chance for icing within the rotor-swept area and near the surface by reducing the production of white-capped waves and the wind-induced tearing of spray off waves. In stable conditions, the mean wake wind speed deficit is largest, reaching $-1.4$ m s$^{-1}$ near the top of the rotor-swept plane, reducing the chance for icing. Because vertical motion is suppressed in stable stratification, winds enhance and flow around and under the wind plant area (Figure 6a), reaching a subtle enhancement near the surface of 0.18 m s$^{-1}$. In unstable stratification, available buoyant turbulence promotes mixing which transports momentum from above the rotor-swept region down to within the wake. The injection of momentum allows wake wind speeds to recover, leaving a smaller maximum averaged wake deficit of $-0.57$ m s$^{-1}$ (Figure 6b). There is no enhancement of wind speeds adjacent to the RIMA block along the cross section in unstable conditions.

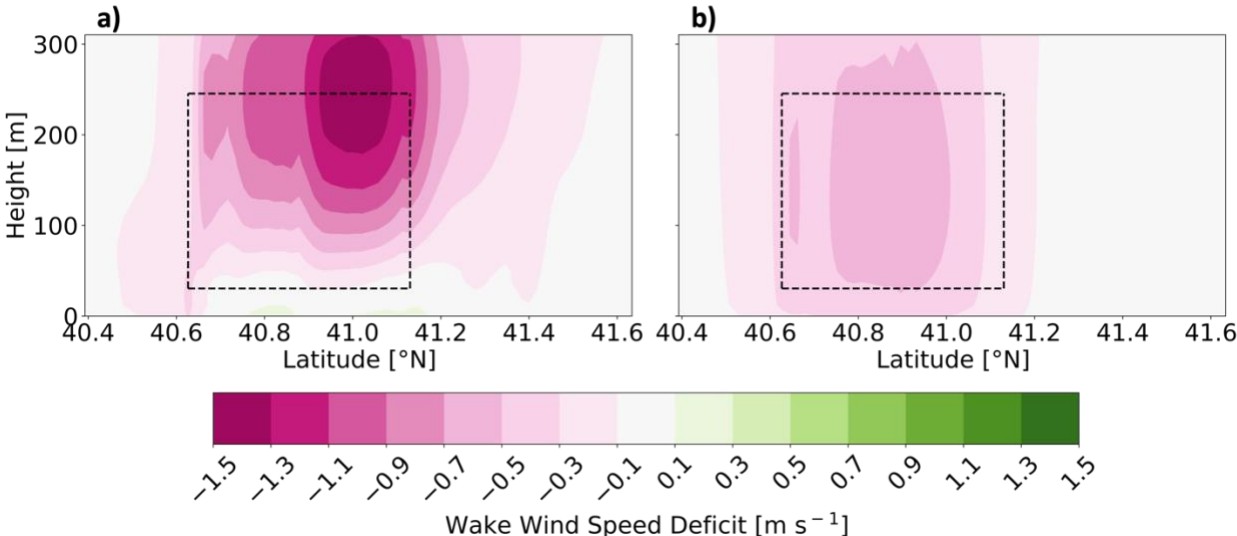


**Figure 6. The mean (WFP-NWF) wind speed difference during (a) stable and (b) unstable stratification, from November**
**2019 to March 2020. The cross section spans the RIMA block of lease areas (Figure 1). Pink contouring indicates a wind**
**speed reduction, and green indicates wind speed enhancement. Dashed lines outline the wind plant area and rotor-swept**
**region. Note the very small enhancement of wind speeds near the surface in stable conditions.**

Despite near-surface cooling, net FSS conditions in WFP occur less often than in NWF when diagnosed using
wind speed, air temperature, and SST criteria because of the wake wind speed reduction. At 10 m, the turbine–
atmosphere interaction alters possible icing conditions the most in February, with a maximum reduction by 15 hours
(Table 3). At 20 m, wind plants cause a reduction by up to 15 hours in January and February. In each case, the
reduction in possible icing conditions is spatially coincident with the wind plant areas (Figure 7). At the 138 m hub
height, the change to the number of FSS hours also maximizes in January and February, with a reduction by 9 hours.

**Table 3. The maximum turbine-induced change in FSS hours by month and height.**

|  | November | December | January | February | March | April |
|---|---|---|---|---|---|---|
| 10 m | 0 | −3 | −14 | −15 | −11 | 0 |
| 20 m | 0 | −4 | −15 | −15 | −12 | 0 |
| 138 m | 0 | −5 | −9 | −9 | −5 | 0 |



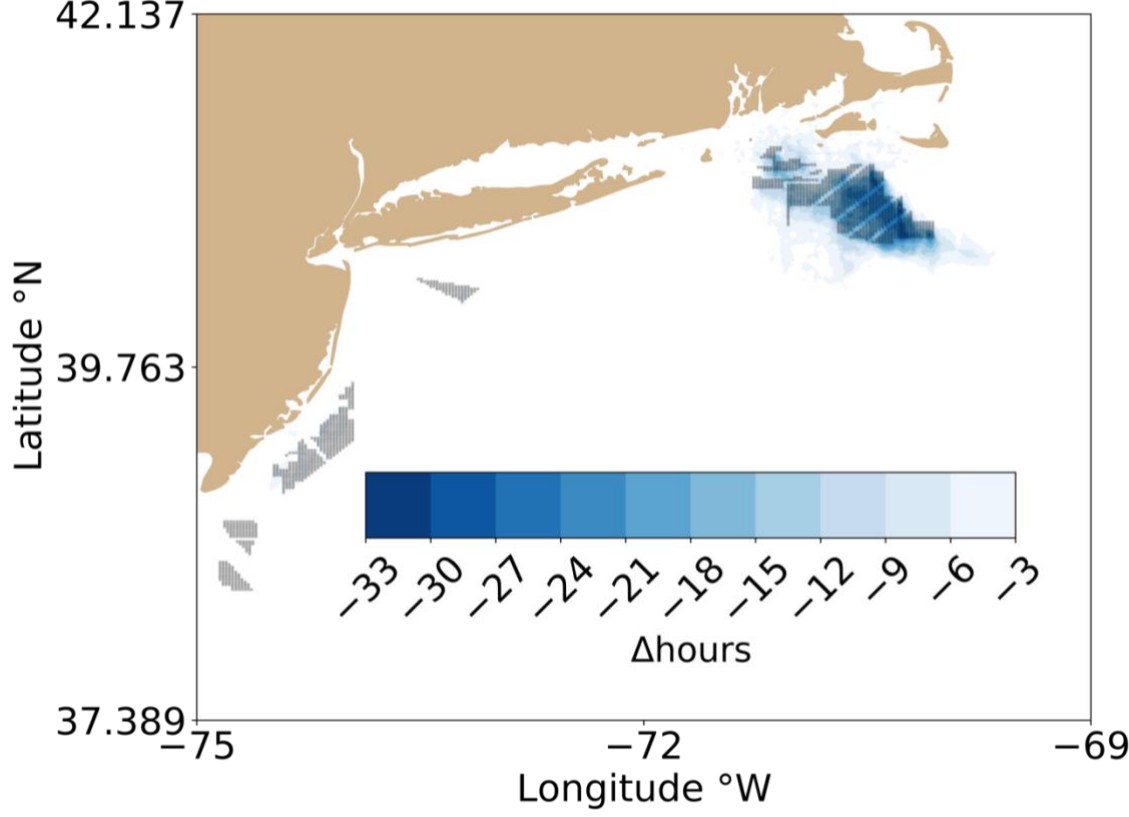


**Figure 7. The (WFP-NWF) change in number of FSS hours at 10 m November 2019 to March 2020. Blue contours**
**indicate a reduction.**

Similarly, the presence of wind turbines has a minimal impact to the number of hours FSS conditions occur by
means of icing PR at the POI. The duration of nonzero PR over the November through March winter period
increases by 3 hours, or from 253 to 256 hours total, at a point centered on the RIMA block. The total duration of
CAO does not change after the installation of wind plants and remains at 202 hours. The total number of events
(seven) does not change in the presence of wind turbines, and all flagged timestamps still cause light icing of less
than 0.7 cm h$^{-1}$.

**4 Conclusions**
Herein, we assess the threat of icing conditions at 10 and 20 m due to freezing sea-spray and at the hub height
due to precipitation and in-cloud icing. The simulation study encompasses the mid-Atlantic Outer Continental Shelf
based on a 21-year WRF dataset from 01 January 2000 to 31 December 2020 and another WRF dataset using year-
long simulations from 01 September 2019 to 31 August 2020. In each case, we focus on the wintertime period from
November through March. We consider the present icing risk from simulations with no wind farms (NOW-23,
NWF) and assess the post-construction adjustments by incorporating the effects of turbines (WFP) in a full buildout
of the wind plant lease areas.

Using an FSS predictability equation (PR), we detect seven events flagged for FSS conditions in NWF with a total duration of 253 hours during the November 2019 to March 2020 period. All times during the period with nonzero icing predictability (PR) contain light ice accumulation of less than 0.7 cm h$^{-1}$, which is typical of the mid-Atlantic bight as assessed from 2000 to 2020. Centered at the RIMA block of lease areas, all seven events have an associated CAO during the 2019–2020 winter. In the NOW-23 dataset from November 2019 to March 2020, eight total events are flagged, and all eight correspond with CAO. Over the 21-year climatology, every CAO event has a corresponding FSS event, although not all FSS events have attendant CAO. Thus, offshore icing conditions may be forecast with reasonable fidelity through accompanying CAO, although other drivers exist. There is strong teleconnection between anomalous arctic sea level pressure sea level pressure and CAO, as 93 % of CAO events in the eastern U.S. contained an antecedent positive arctic sea level pressure anomaly a week in advance (Vavrus et al., 2006).

The number of FSS hours exhibit spatial variability, as assessed using our detection criteria of low air sea surface temperatures and strong winds . The hazards intensify toward higher latitudes where air and sea temperatures are colder and wind speeds are faster, near the land surface where cold air advects offshore, and by Nantucket and the Long Island Sound where SSTs are colder. Icing conditions at the hub height, as assessed by low air temperatures and precipitation or saturated air, are more frequent. The icing hazard is greatest during January when wind speeds are fast and temperatures are cold. At 10 m in January, favorable conditions for icing occur up to 67 hours. At 20 m in January, the duration of icing conditions is similar at 68 hours. Finally, at the hub height, icing conditions occur for up to 119 hours east of Cape Cod. Overall, the 2019–2020 winter period is the mildest winter when considering the 21-year climatology. Although the 2019–2020 winter season has the fewest number of freezing sea spray hours, all winters contain light ice accumulation rates of 0.7 cm h$^{-1}$.

The introduction of large wind plants makes a small impact on the icing risk within the wind plant clusters. In wintertime unstable conditions, which occur 64 % of the time from November 2019 through March 2020, wind turbines introduce a mean near-surface cooling effect. Despite the enhanced freezing risk from supplementary cooling, slower wind speeds in the wake mitigate the icing hazard. A mean reduction in wind speeds within wakes reaches up to $-0.57$ m s$^{-1}$ in unstable stratification with a mean introduction of cooler air up to $-0.041$ K. As assessed using wind speed, air temperature, and SST criteria, the change in FSS risk over the 2019–2020 wintertime period is a net reduction, by only 15 hours at both 10 and 20 m. The alleviation by slower wind speeds is largest within the RIMA block of wind plants which contains the greatest number of turbines and the greatest number of FSS hours relative to other wind energy areas. When assessed using PR centered on the RIMA block, the number of icing hours increases by 3 with no change to CAO hours. Although the 2019 through 2020 winter period is the mildest winter, and thus not representative of the 21-year climatology of FSS conditions, this period captures well the post-construction effects of wind plants. We note that such effects may be more significant during during harsher winters.

433

Future OCS winter storm frequency may differ due to climate change. For instance, warming Arctic
temperatures, which reduce the meridional geopotential height gradient between the Arctic and midlatitudes, can
weaken the jet stream. Slower zonal winds and more pronounced Rossby waves amplify the transport of extreme
winter weather to the midlatitudes (Cohen et al., 2020). Future East Coast storm activity and temperature may
experience modulations based on large-scale teleconnections such as El Niño and the North Atlantic Oscillation
(Hall and Booth, 2017). Further, Arctic amplification may increase the strength of teleconnection found between
positive Arctic sea level pressure anomalies and CAO (Vavrus et al., 2006).

Finally, we assume that sea spray provides a consistent moisture flux at 10 and 20 m during fast wind
conditions, that the droplet size of spray is homogeneous, and that the number distribution by height is constant. The
impingement of waves onto offshore structures provides a larger source of moisture than wind-generated spray that
is dependent on the wave height and wave period. Future studies may benefit from coupling WRF with wave
models, such as Wave Watch III (Tolman et al., 2019) and Simulating WAves Nearshore (SWAN Team, 2020) for
precise modeling of wave characteristics and current dynamics, such as stratified cold pooling around Cape Cod.
New satellite methods are being developed to quantify occurrences of freezing sea spray (Line et al., 2022), and
future developments could compare the FSS criteria to satellite observations of FSS.

**5 Code and data availability**

The dataset and files that support this work are publicly available. The ERA5 initial and boundary conditions can be
downloaded from the ECMWF Climate Data Store at https://cds.climate.copernicus.eu/cdsapp#!/dataset/reanalysis-
era5-pressure-levels?tab=form. Shapefiles including the bounds for wind energy lease areas are at
https://www.boem.gov/renewable-energy/mapping-and-data/renewable-energy-gis-data. Wind turbine coordinates
and their power and thrust curves are provided at https://zenodo.org/record/7374283#.Y4YZxC-B1KM. WRF
namelists for NWF and WFP simulations may be acquired from https://zenodo.org/record/7374239#.Y4YaOy-
B1KM. The NOW-23 simulation output data are available in HDF5 format at https://doi.org/10.25984/1821404.

**6 Author contributions**

Conceptualization: JKL. Resources: MO, NB. Methodology: DR, JKL. Software: DR. Formal analysis and
visualization: DR. Investigation: DR and JKL. Writing – original draft: DR and JKL. Writing – review and editing:
all co-authors. Supervision: JKL.

**7 Competing interests**

At least one of the (co-)authors is a member of the editorial board of Wind Energy Science. Furthermore, Mike
Optis co-authored the submitted manuscript while an employee of the National Renewable Energy Laboratory. He
has since founded Veer Renewables, which recently released a wind modeling product, WakeMap, which is based
on a similar numerical weather prediction modeling framework as the one described in this manuscript. Data from
WakeMap is sold to wind energy stakeholders for profit. Public content on WakeMap include a website (htt

J.Mps://veer.eco/wakemap/), a white paper (https://veer.eco/wp-
content/uploads/2023/02/WakeMap_White_Paper_Veer_Renewables.pdf) and several LinkedIn posts promoting
WakeMap. Mike Optis is the founder and president of Veer Renewables, a for-profit consulting company. Mike
Optis is a shareholder of Veer Renewables and owns 92 % of its stock.
**8 Acknowledgements**
This work utilized the Alpine high-performance computing resource at the University of Colorado Boulder. Alpine
is jointly funded by the University of Colorado Boulder, the University of Colorado Anschutz, and Colorado State
University. Data storage supported by the University of Colorado Boulder 'PetaLibrary' A portion of this research
was performed using computational resources sponsored by the DOE's Office of Energy Efficiency and Renewable
Energy and located at NREL. This work was authored in part by the National Renewable Energy Laboratory,
operated by Alliance for Sustainable Energy, LLC, for the US Department of Energy (DOE) under contract no. DE-
AC36-08GO28308. Funding was provided by the US Department of Energy Office of Energy Efficiency and
Renewable Energy Wind Energy Technologies Office. Support for the work was also provided by the National
Offshore Wind Research and Development Consortium under agreement no. CRD-19-16351. The views expressed
in the article do not necessarily represent the views of the DOE or the US Government. The US Government retains
and the publisher, by accepting the article for publication, acknowledges that the US Government retains a
nonexclusive, paid-up, irrevocable, worldwide license to publish or reproduce the published form of this work, or
allow others to do so, for US Government purposes.
The authors wish to thank Louis Bowers and Sarah McElman for their questions that led to this line of inquiry.
**9 Appendices**
**Appendix A**

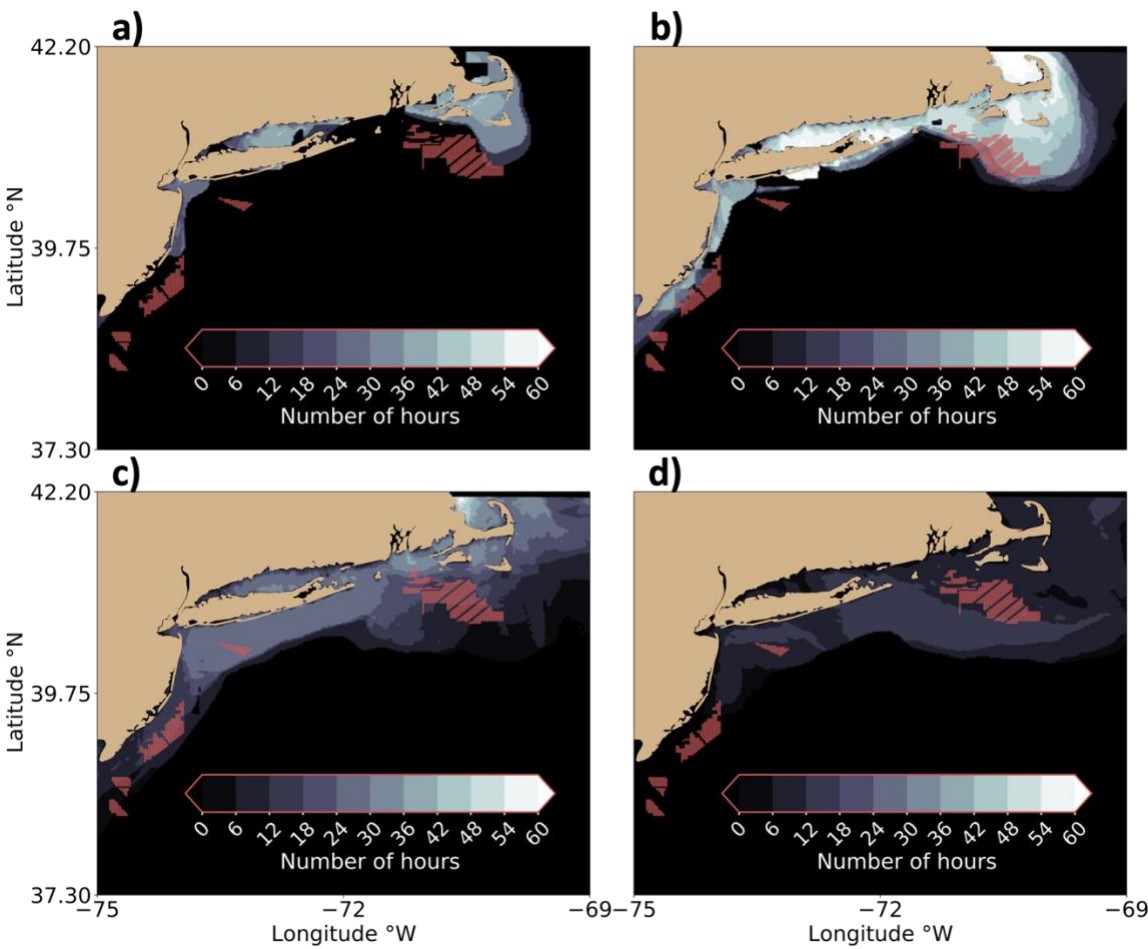


**Figure. A1. The number of freezing hours at 10 m during (a) December 2019, (b) January 2020, (c) February 2020, and (d) March 2020. Lighter contouring indicates higher percentages. Red dots indicate turbine locations.**


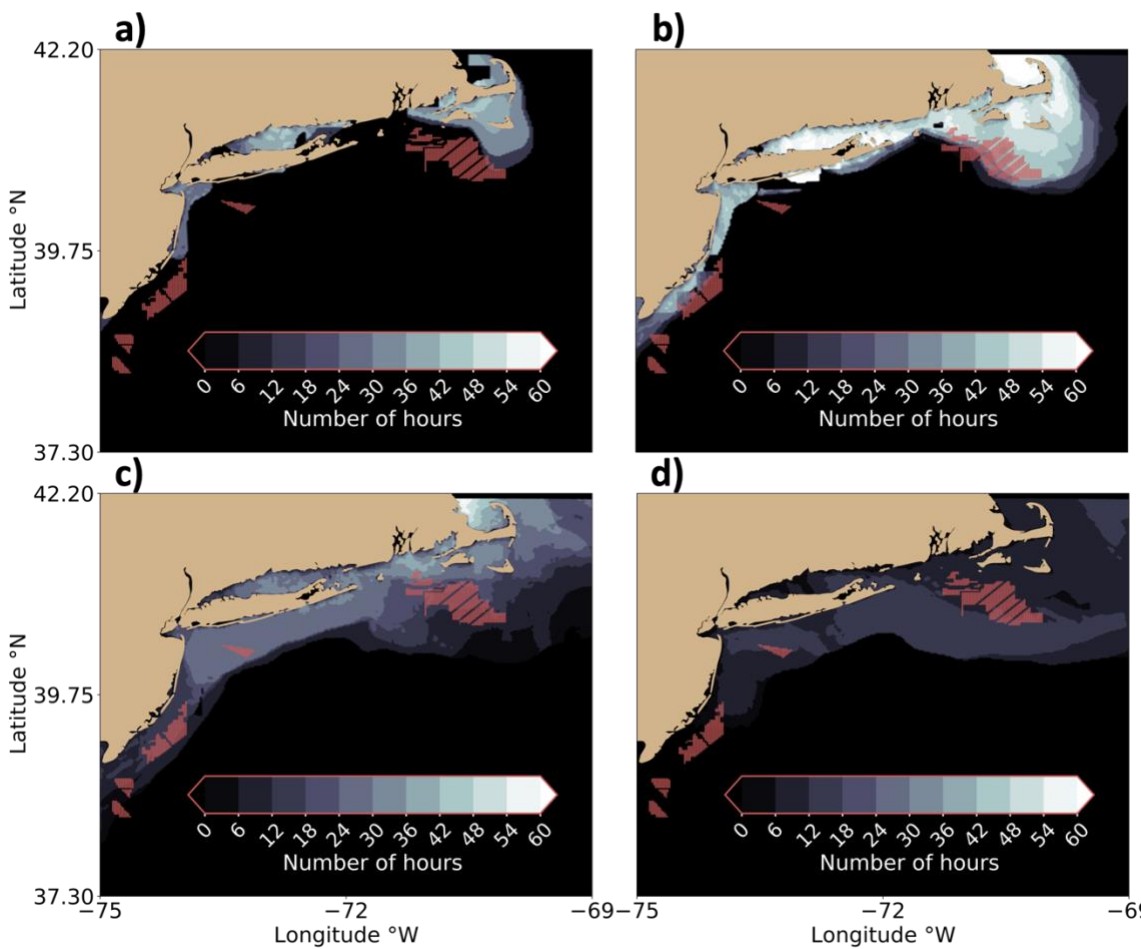


**Figure A2. The number of freezing hours at 20 m during (a) December 2019, (b) January 2020, (c) February 2020, and (d) March 2020. Lighter contouring indicates higher percentages. Red dots indicate turbine locations.**


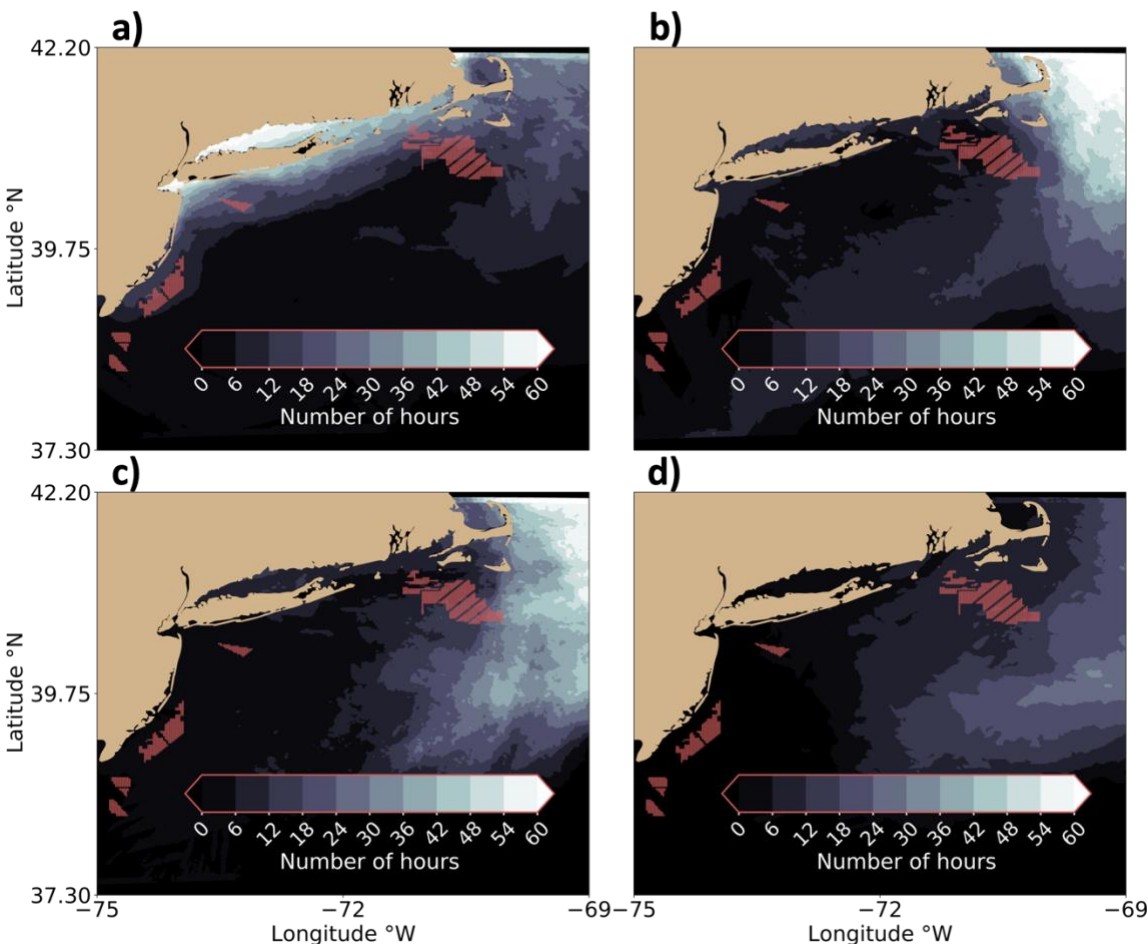


Figure A3. The number of freezing hours at hub height during (a) December 2019, (b) January 2020, (c) February 2020, and (d) March 2020. Lighter contouring indicates higher percentages. Note the color scheme is different from Supplementary Figs. 1 and 2. Red dots indicate turbine locations.



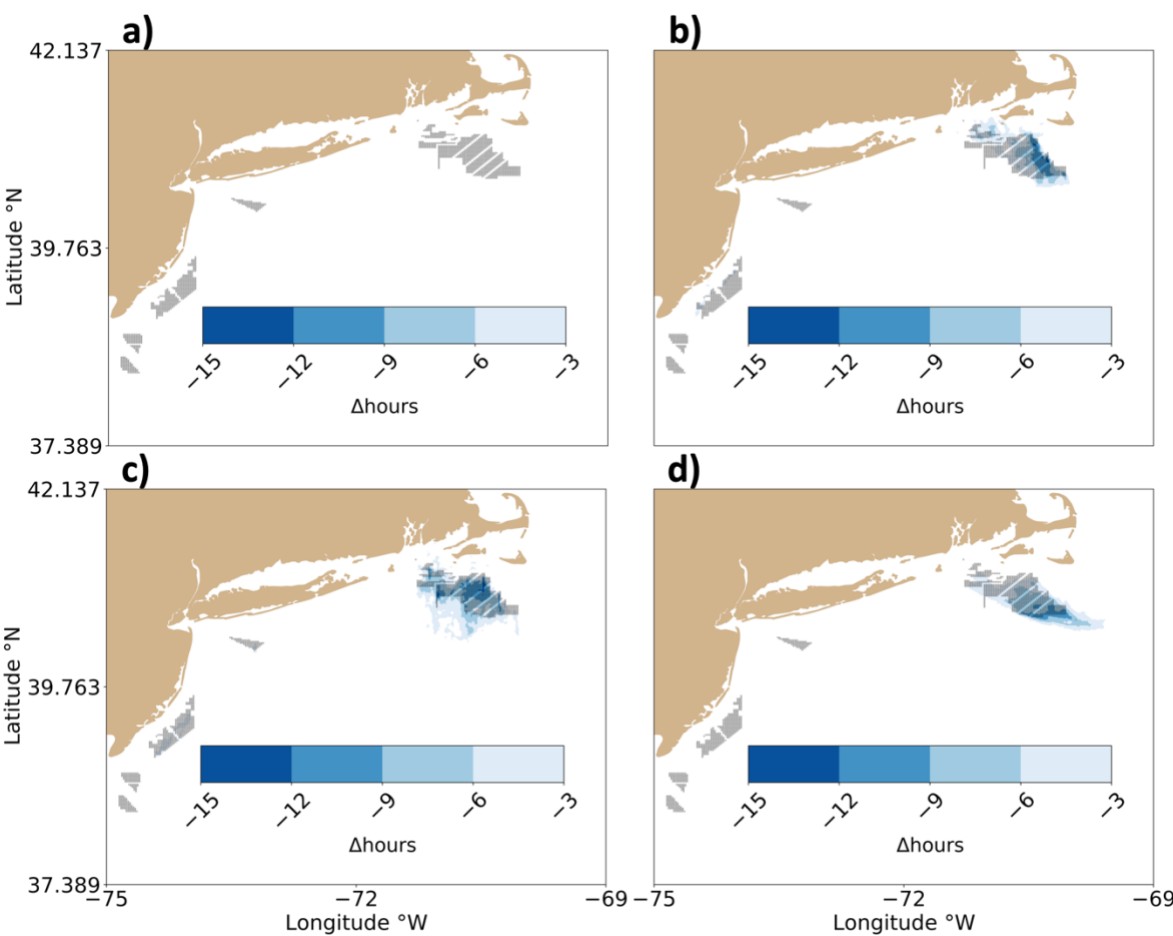


Figure A4. The (WFP-NWF) difference in freezing hours at 10 m during (a) December 2019, (b) January 2020, (c) February 2020, and (d) March 2020. Blue contouring indicates fewer hours. Gray dots indicate turbine locations.


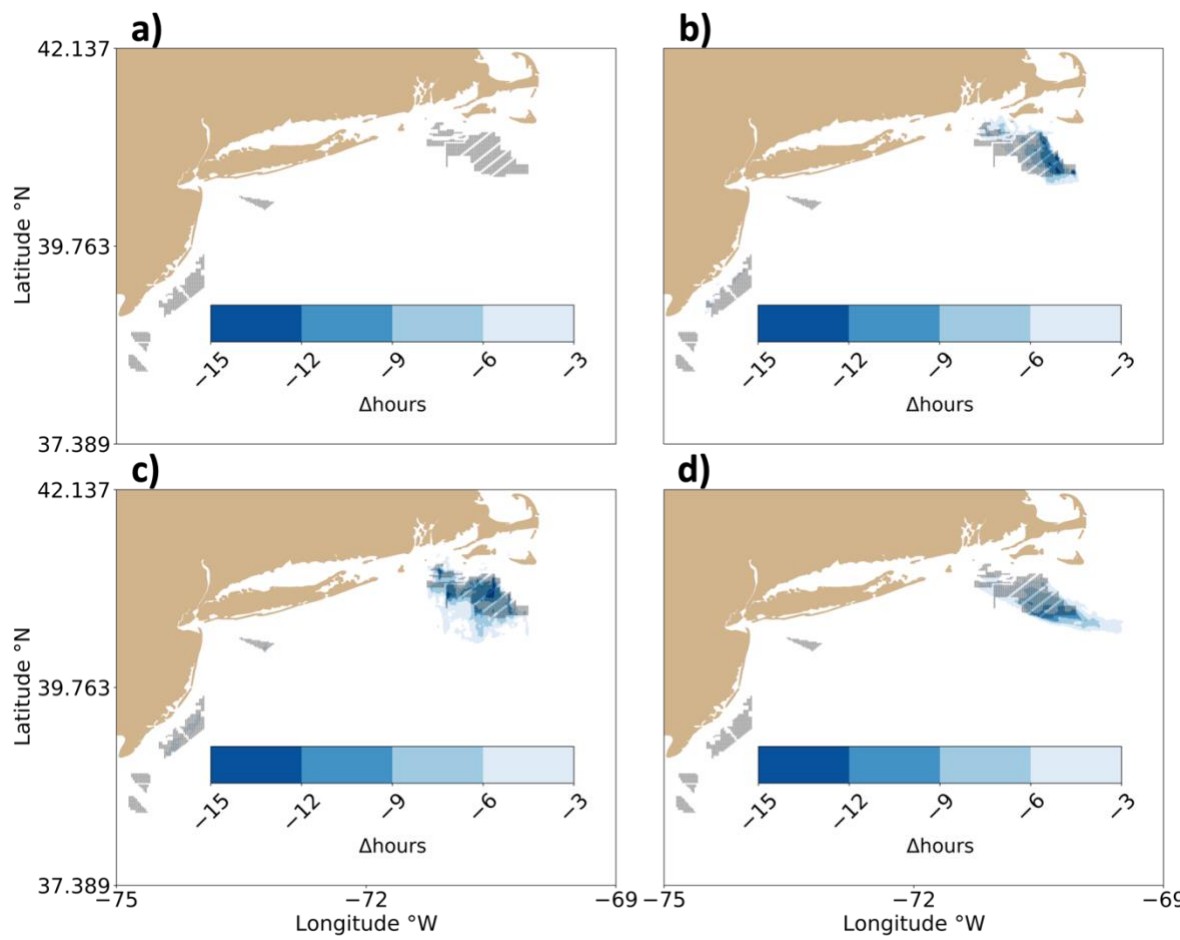


**Figure A5. The (WFP_0-NWF) difference in freezing hours at 20 m during (a) December 2019, (b) January 2020, (c)**
**February 2020, and (d) March 2020. Blue contouring indicates fewer hours. Gray dots indicate turbine locations.**

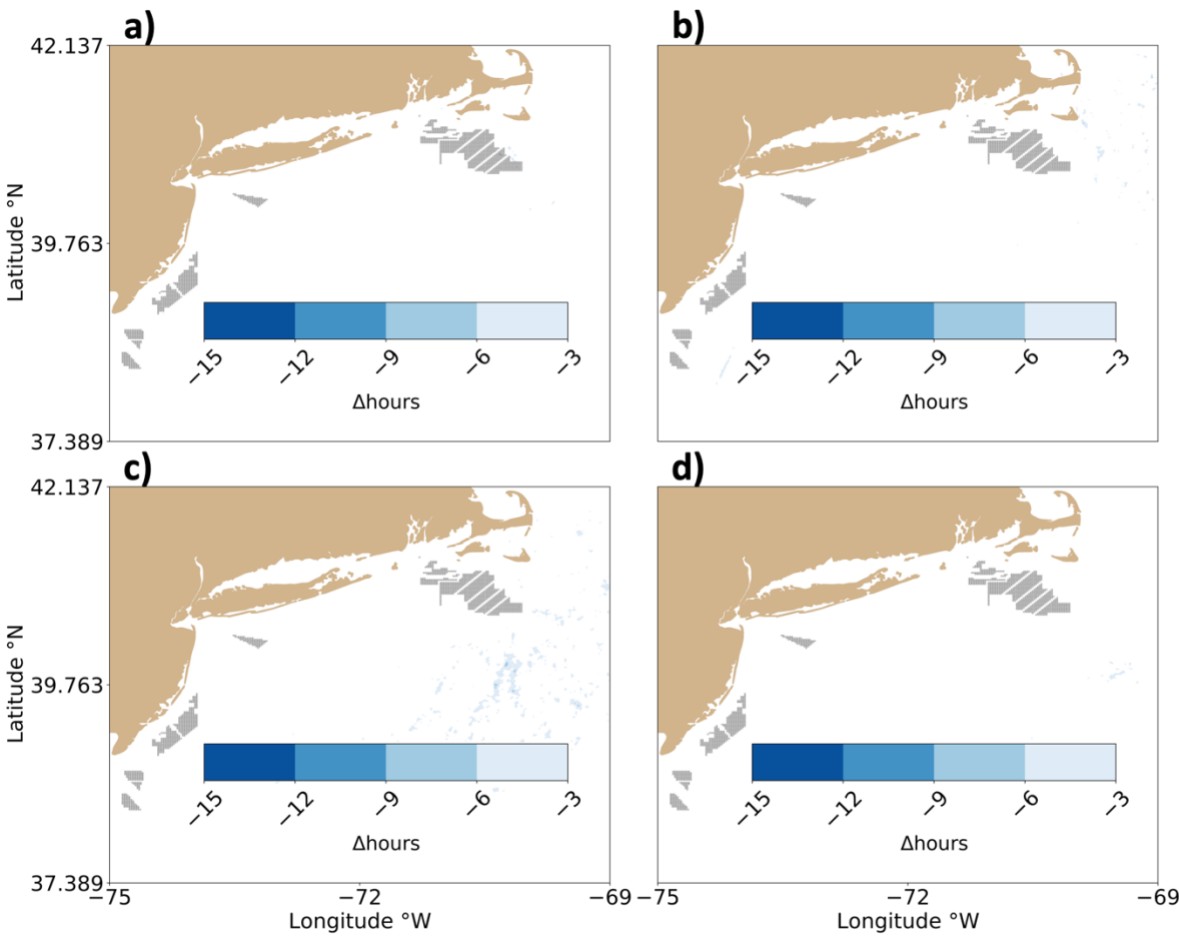


**Figure A6. The (WFP-NWF) difference in freezing hours at the hub height during (a) December 2019, (b) January 2020,**
**(c) February 2020, and (d) March 2020. Blue contouring indicates fewer hours. Gray dots indicate turbine locations.**
**Appendix B**
As discussed in Section 2.3, we detect FSS conditions using common thresholds for the meteorological
conditions (Guest and Luke, 2005; Dehghani-Sanij et al., 2017; Line et al., 2022). These criteria require strong wind
speeds greater than 9 m s$^{-1}$, cold air temperatures below −1.7° C, and cold SSTs less than 7° C. As reviewed by
Dehghani-Sanij et al., (2017), FSS conditions are promising when the air temperature is below either −1.7° C or −2°
C to account for the lower freezing point of saline ocean water; the salt content of which determines this threshold.
Although SST thresholds of 5° C or 7° C are prevalent, a threshold up to 8.9° C has been used (U.S. Navy, 1988).
As such, we quantify some of the uncertainty by calculating the number of hours that FSS conditions occur using
conservative thresholds, which produce fewer icing hours (FEWER), and liberal thresholds, which promote more
icing hours (MORE) (Table B1). As there is wider agreement regarding the wind speed threshold (Dehghani-Sanij et
al., 2017; Guest and Luke, 2005; Line et al., 2022; Ross and Cardone, 1974; Monahan et al., 1983; Monahan and
MacNiocaill, 1986), we hold it constant. Due to computational constraints, we only assess the number of icing hours
throughout the domain at 10 m and during January 2020 because it has the greatest number of icing hours.

**Table B1. Icing detection criteria by sensitivity analysis type.**

| Acronym | Air temperature | Sea surface temperature | Wind speed |
|---------|-----------------|-------------------------|------------|
| **FEWER** | $<-2°$ C | $<5°$ C | $>9$ m s$^{-1}$ |
| **MORE** | $<-1.7°$ C | $<8.9°$ C | $>9$ m s$^{-1}$ |

As expected, more conservative thresholds produce fewer FSS hours and vice versa (Fig. B1a,b,c). In FEWER, the meteorological conditions conducive to icing maximize at 60 hours. Using more liberal criteria in MORE, the maximum number of hours increases to 67. Despite the small change in the maximum number of hours FSS occurs, the regional variation is large; the area covered by icing conditions increases from 8,924 km$^2$ to 135,244 km$^2$ from FEWER to MORE, or roughly 15 times greater than FEWER, or 2.2 times greater than our production set of criteria. Regional variability follows SST patterns and only occurs in FEWER where the SST is relatively cold in the Long Island Sound and Nantucket Sound (Table B1b), as discussed previously.

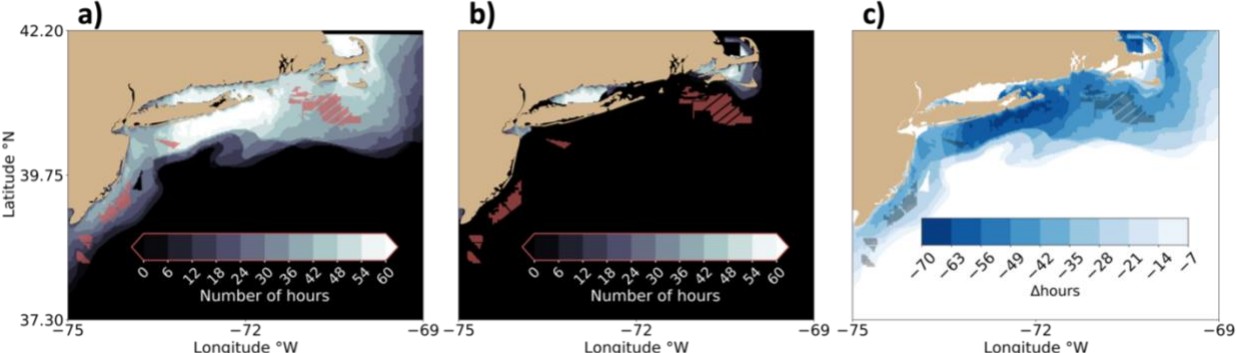

**Fig. B1. The number of hours FSS conditions occur during January 2020 at 10 m in NWF using thresholds for (a) FEWER, (b) MORE, and (c) the (FEWER-MORE) difference. Lighter contouring indicates more freezing hours in (a) and (b). Darker blues represent a larger reduction in number of hours in (c). Turbine locations are shown as red dots in (a) and (b) and as black dots in (c).**

**Appendix C**

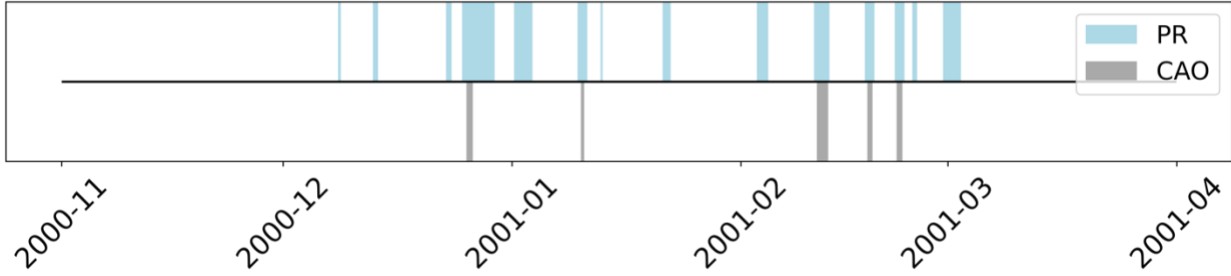

**Fig. C1. Time series of CAO and FSS events from November 2000 to April 2001. Light-blue shading indicates the duration of nonzero PR and gray shading indicates the duration of detected CAO from NOW-23.**

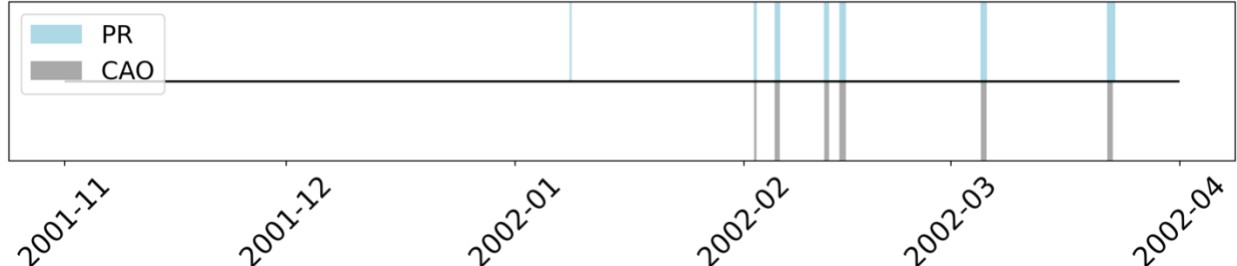

Fig. C2. Time series of CAO and FSS events from November 2001 to April 2002. Light-blue shading indicates the duration of nonzero PR and gray shading indicates the duration of detected CAO from NOW-23.

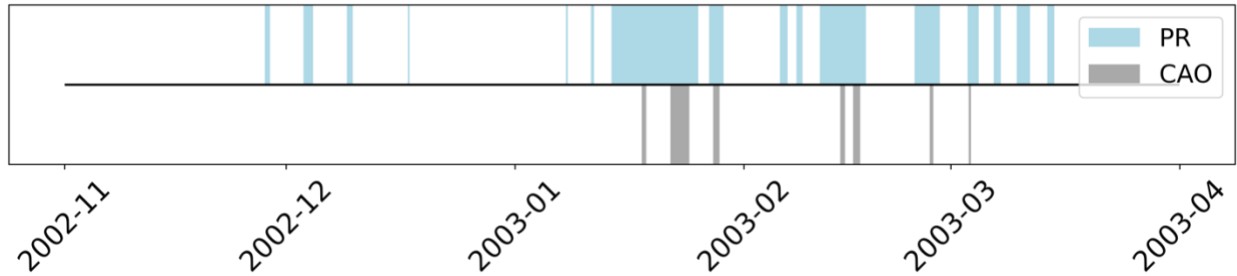

Fig. C3. Time series of CAO and FSS events from November 2002 to April 2003. Light-blue shading indicates the duration of nonzero PR and gray shading indicates the duration of detected CAO from NOW-23.

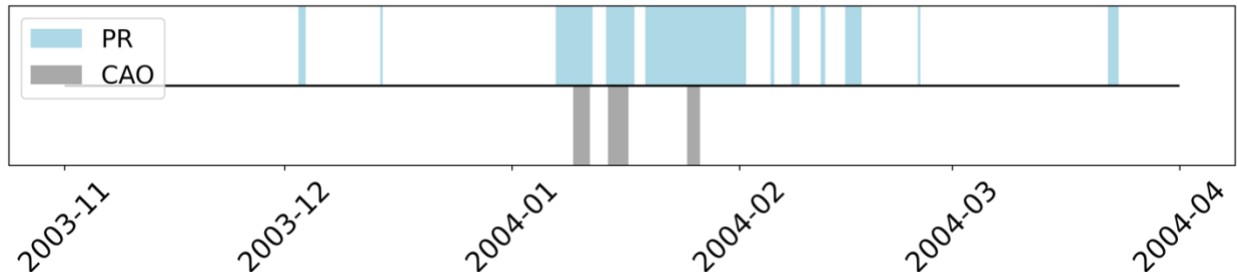

Fig. C4. Time series of CAO and FSS events from November 2003 to April 2004. Light-blue shading indicates the duration of nonzero PR and gray shading indicates the duration of detected CAO from NOW-23.

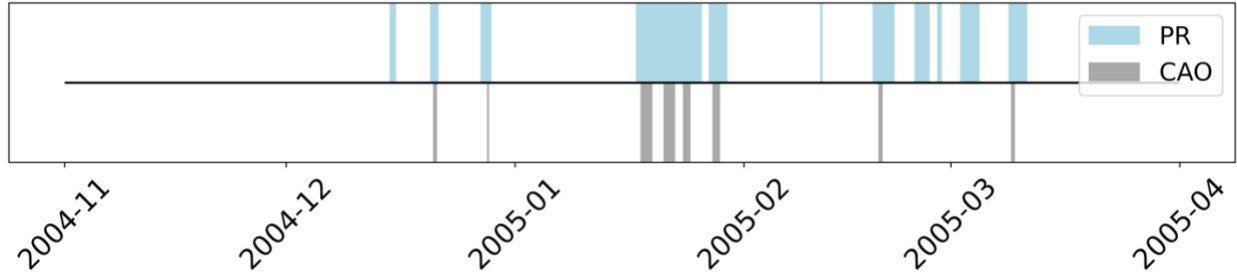

Fig. C5. Time series of CAO and FSS events from November 2004 to April 2005. Light-blue shading indicates the duration of nonzero PR and gray shading indicates the duration of detected CAO from NOW-23.

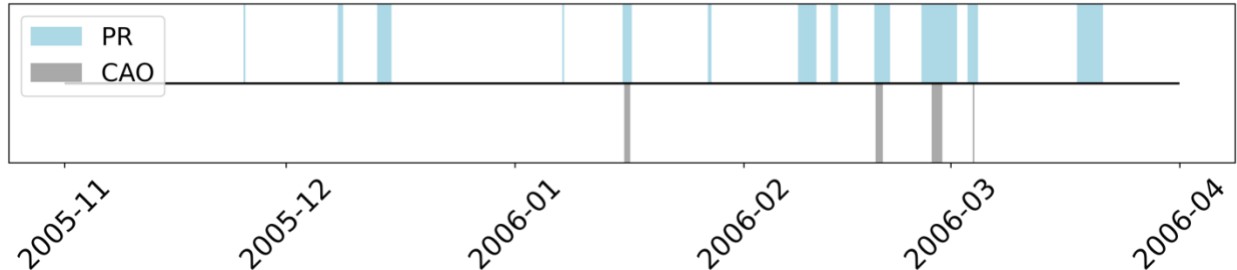

**Fig. C6.** Time series of CAO and FSS events from November 2005 to April 2006. Light-blue shading indicates the duration of nonzero PR and gray shading indicates the duration of detected CAO from NOW-23.

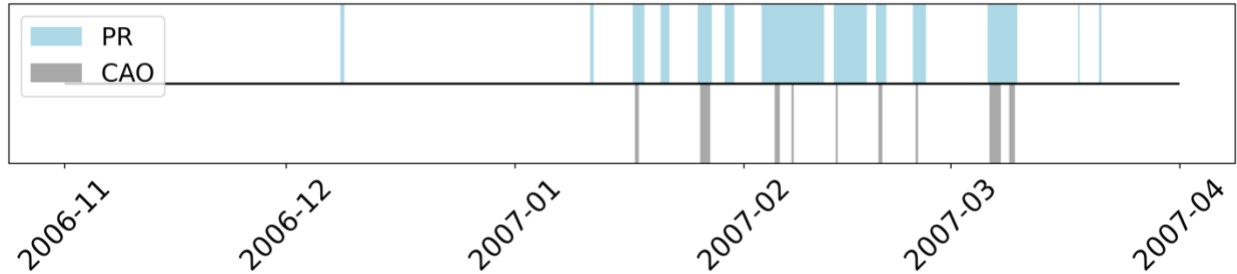

**Fig. C7.** Time series of CAO and FSS events from November 2006 to April 2007. Light-blue shading indicates the duration of nonzero PR and gray shading indicates the duration of detected CAO from NOW-23.

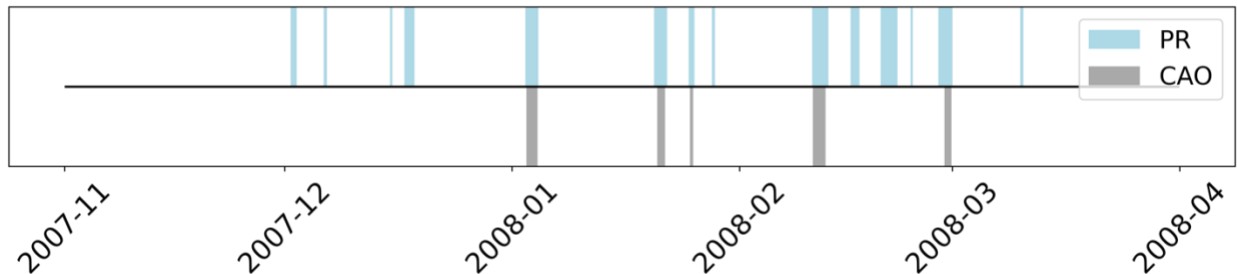

**Fig. C8.** Time series of CAO and FSS events from November 2007 to April 2008. Light-blue shading indicates the duration of nonzero PR and gray shading indicates the duration of detected CAO from NOW-23.

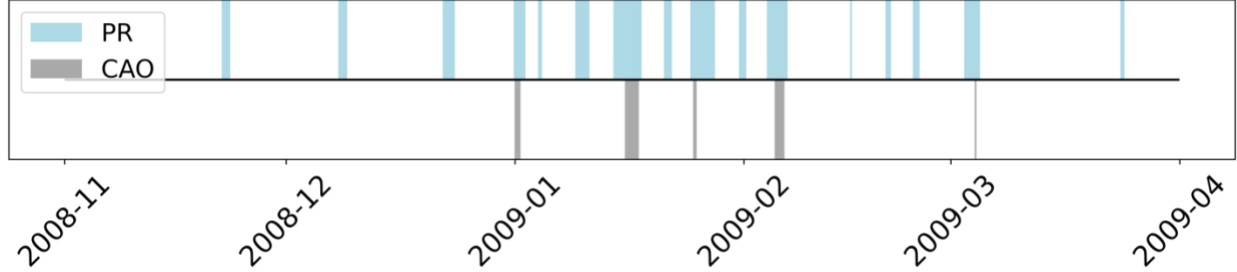

**Fig. C9.** Time series of CAO and FSS events from November 2008 to April 2009. Light-blue shading indicates the duration of nonzero PR and gray shading indicates the duration of detected CAO from NOW-23.

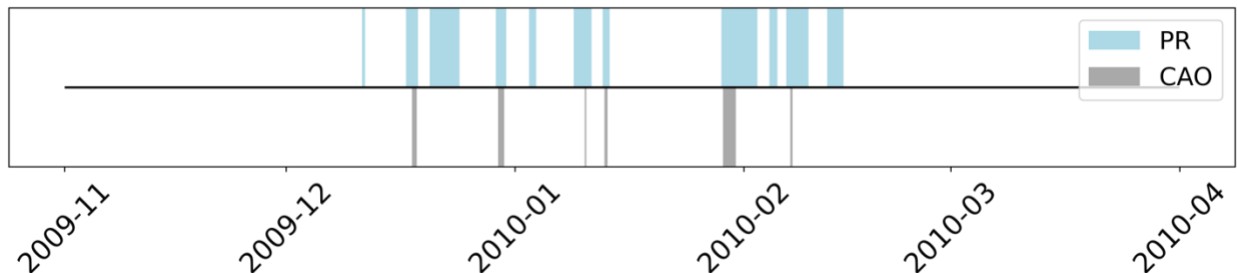

Fig. C10. Time series of CAO and FSS events from November 2009 to April 2010. Light-blue shading indicates the duration of nonzero PR and gray shading indicates the duration of detected CAO from NOW-23.

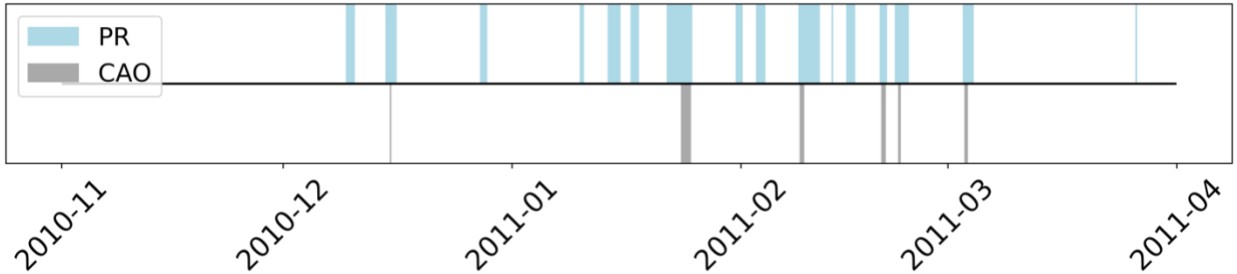

Fig. C11. Time series of CAO and FSS events from November 2010 to April 2011. Light-blue shading indicates the duration of nonzero PR and gray shading indicates the duration of detected CAO from NOW-23.

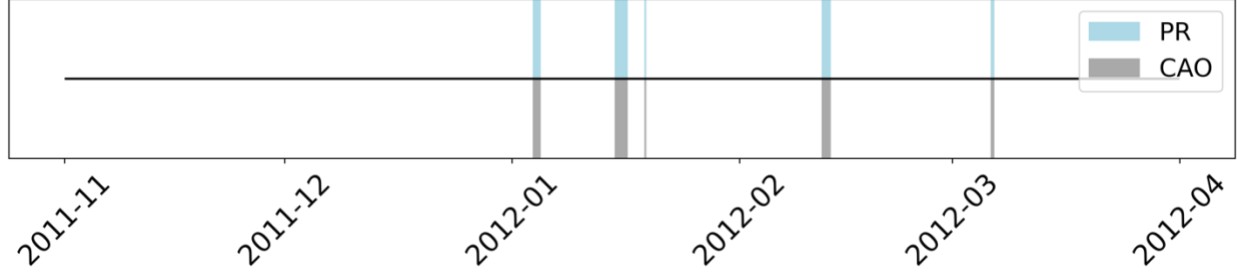

Fig. C12. Time series of CAO and FSS events from November 2011 to April 2012. Light-blue shading indicates the duration of nonzero PR and gray shading indicates the duration of detected CAO from NOW-23.

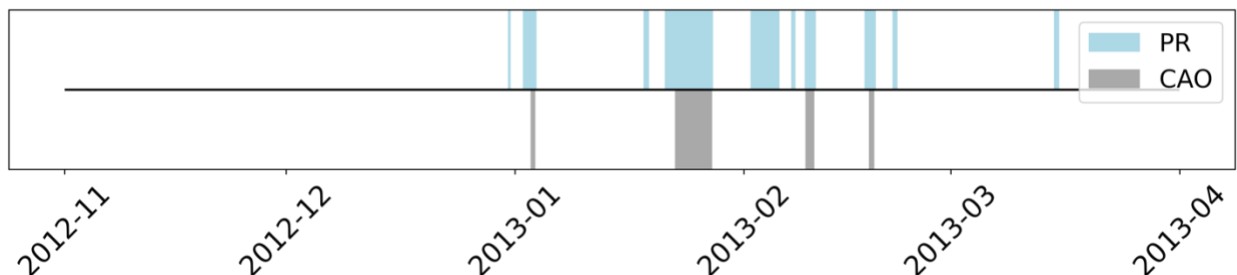

Fig. C13. Time series of CAO and FSS events from November 2012 to April 2013. Light-blue shading indicates the duration of nonzero PR and gray shading indicates the duration of detected CAO from NOW-23.

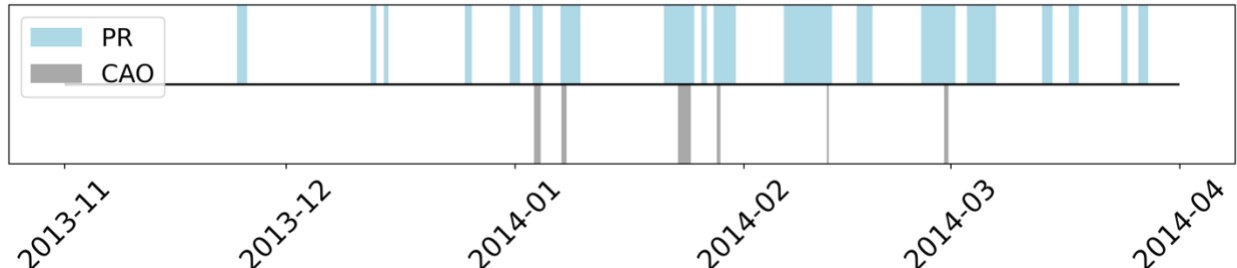

**Fig. C14. Time series of CAO and FSS events from November 2013 to April 2014. Light-blue shading indicates the**
**duration of nonzero PR and gray shading indicates the duration of detected CAO from NOW-23.**

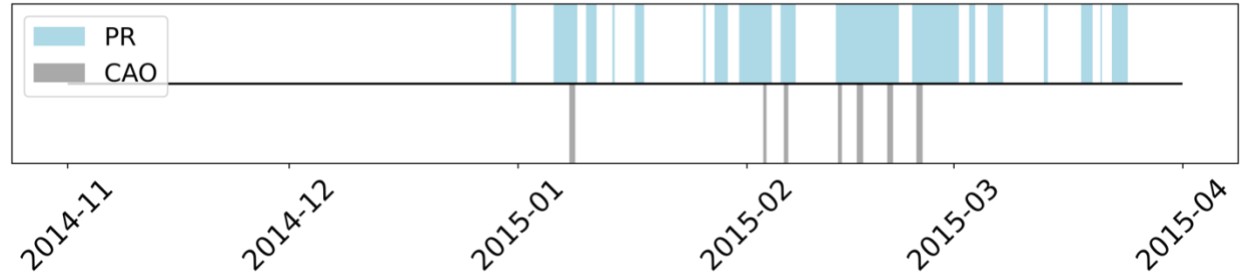

**Fig. C15. Time series of CAO and FSS events from November 2014 to April 2015. Light-blue shading indicates the**
**duration of nonzero PR and gray shading indicates the duration of detected CAO from NOW-23.**

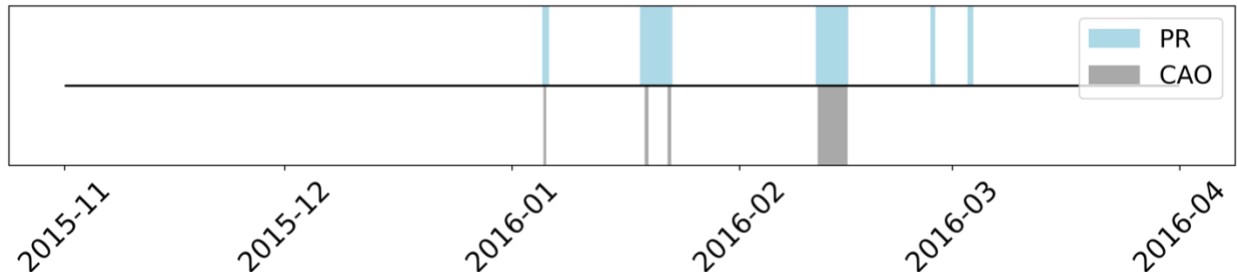

**Fig. C16. Time series of CAO and FSS events from November 2015 to April 2016. Light-blue shading indicates the**
**duration of nonzero PR and gray shading indicates the duration of detected CAO from NOW-23.**

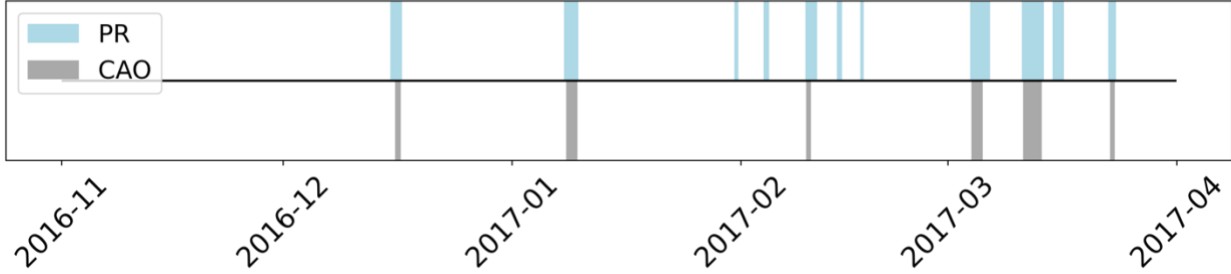

**Fig. C17. Time series of CAO and FSS events from November 2016 to April 2017. Light-blue shading indicates the**
**duration of nonzero PR and gray shading indicates the duration of detected CAO from NOW-23.**

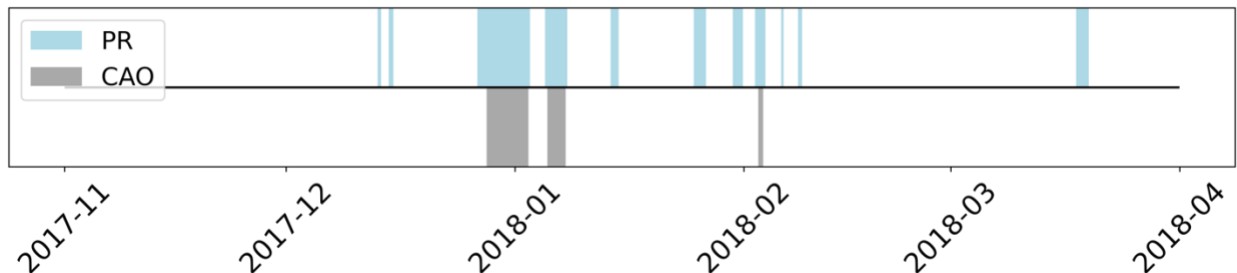

**Fig. C18. Time series of CAO and FSS events from November 2017 to April 2018. Light-blue shading indicates the**
**duration of nonzero PR and gray shading indicates the duration of detected CAO from NOW-23.**

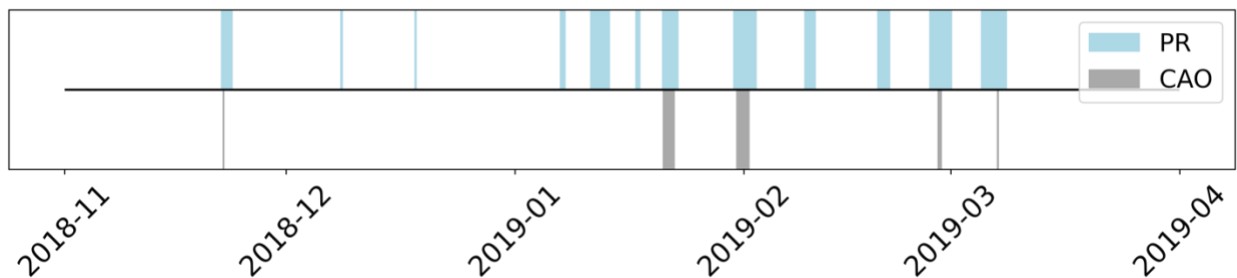

**Fig. C19. Time series of CAO and FSS events from November 2018 to April 2019. Light-blue shading indicates the**
**duration of nonzero PR and gray shading indicates the duration of detected CAO from NOW-23.**

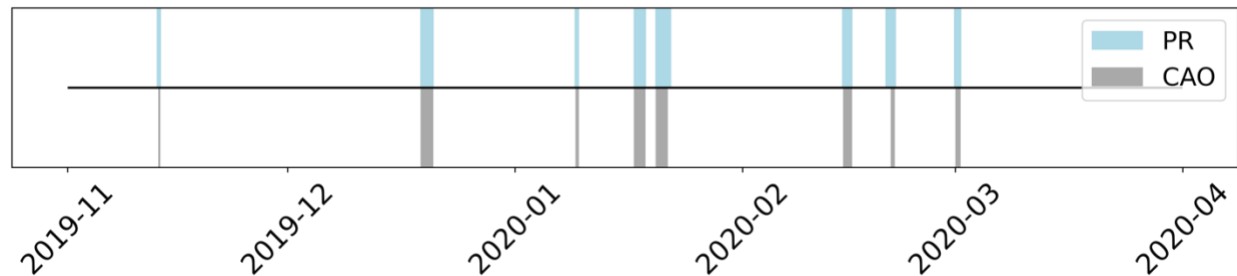

**Fig. C20. Time series of CAO and FSS events from November 2019 to April 2020. Light-blue shading indicates the**
**duration of nonzero PR and gray shading indicates the duration of detected CAO from NOW-23.**

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
