# Peer review of "The Effects of Wind Farm Wakes on Freezing Sea Spray in the Mid-Atlantic Offshore Wind Energy Areas"

_Wind Energy Science, 2024_

## Author Comment (AC1)

Reviewer comments appear in **black** and author responses in **blue**.

**Reviewer 1**

The paper addresses the effect of a large wind farm cluster on the frequency and magnitude of sea spray freezing events. The conclusion is that the effect of even a very large cluster, and in an environment prone to these events, is minimal. The paper thus ends the scientific debate on this topic.
We thank the reviewer for reading our article thoroughly and for providing feedback to improve this work.

There is potential for improvement. Inconsistency and inaccuracy in how the authors present the hypotheses, scope, methods, and results, are plentiful and should be corrected. For example, it has evaded the senior authors' detection how "freezing" and "icing" are used interchangeably due to apparent lack of understanding of the two separate phenomena. Please see the specific comments below.
Given the paper's light scientific weight, it may be published at the editor's discretion, after these comments are taken into account and the paper revised accordingly. Therefore the required revision is classified as "minor".
Thank you for providing this feedback. We have addressed our interchangeable use of freezing and icing; freezing requires temperatures below 0°C and icing requires freezing conditions in addition to a moisture source.

Abstract:
15-16, 21 "liquid precipitation" is not the most common cause of ice accretion. Liquid cloud particles cause most of the icing which causes the aerodynamic degradation of the wind turbine blades, which in turn causes "extra load and fatigue" and loss of production (not mentioned in the Abstract - please add). Icing on the blades is usually not called "freezing" and I suspect that the authors refer to for example freezing of the sea spray on the service boats and access platforms. This inconsistency repeats throughout the paper - please fix it.
Thank you for pointing this out. We have changed this sentence to: "Ice accretion on turbine rotors and blades occurs from precipitation and in-cloud icing at temperatures below freezing. Ice accretion induces load and fatigue on mechanical parts which reduces blade performance and power production."

24 "not all freezing events are cold enough to signify a cold air outbreak" ... unclear what the meaning of this statement is. Cold air outbreak is mentioned several times in the paper, unnecessarily.
We have removed this sentence from the abstract because it is not a vital finding. However, we do think that it is necessary to emphasize the role of cold air outbreaks

in FSS because we see that all cold air outbreaks cause FSS, and so FSS can be predicted.

26 The wakes are said to "mitigate the chance for freezing". Given that the effect of wakes to wind farm performance is significant and more important than freezing, this statement should perhaps be revised.
Our goal in this work is to characterize the effect of wakes on the number of icing hours, not to wind farm performance. We have modified this sentence to correctly refer to "icing" instead of "freezing". The significant impacts of wakes on wind farm performance is a larger topic addressed in our other work, Rosencrans et al. 2024, and cited herein.

Introduction:
34 Is the White House a credible scientific source/reference?
We now point to a similar metric from an NREL technical report: "In the U.S., offshore capacity targets are approaching 40 GW by 2040 (Musial et al., 2022)."

39 The effect of ice on energy production is mentioned here. Please include it into the Abstract (see my comment above)
We have clarified that the reduction of rotor aerodynamic efficiency causes a reduction of power production in the abstract.

42 If the rotation stops entirely, then one would expect the power production to reduce by 100%, not just up to 80%.
We have modified the sentence to report the total power losses over the full icing event, which featured reduced aerodynamic efficiency, slower blade rotation, and, at times (but not at all times), full stoppage: "One study found that excessive icing induced a power loss of 63 % for a single turbine over a 51-h icing event (Gao and Hu, 2021)."

43, 45 The reference is irrelevant, why are just two case studies selected? Please consider together with the references in 49 (Martini et al. ...). The turbine blade icing effect is indeed well studied, and many of the 542 references in the IEA Task 19 technical report (https://iea-wind.org/wp-content/uploads/2021/09/Lehtomaki-et-al.-2018-Available-Technologies-for-Wind-Energy-in-Cold-Climates-report-2-nd-edition-2018.pdf) would be more appropriate. Generally, the reference should point to the earliest appropriate publication, not to a random one.
Thank you for bringing up this point. We have cited a review paper on wind turbine ice-induced power losses (Contreras Montoya et al., 2022) which points to a large body of work surrounding the subject, and thus have no need to cite two case-study scenarios. We have removed the sentences featuring the two case study scenarios

and have referenced IEA Task 19 to provide a wide-reaching springboard for the readers.

47 Is there evidence that the winds are faster in cold air outbreaks, than in for example warm air outbreaks?
We are not comparing cold air outbreaks (a specific meteorological occurrence, as discussed in (Atkinson and Wu Zhang, 1996; Geerts et al., 2022; Vavrus et al., 2006) to warm air outbreaks (which do not have a specific meteorological label). We have clarified that strong winds occur during cold fronts relative to typical wind speeds: "Faster winds during cold front passages can enhance wind-energy supply during high-load cold-weather events, although, following frontal passages, the combination of cold temperatures and slow wind speeds may pose severe challenges for utility grid planners (Novacheck et al., 2021)."

69 Minor note: turbulence does not transport temperature, but heat.
This sentence has been rephrased as follows: "Enhanced turbulence caused by spinning blades transports heat from aloft to lower altitudes within the rotor-swept region or near the surface".

77 It is perhaps interesting, but not "crucial" to understand how large scale deployment of wind farms will modify freezing events.
This sentence has been modified to reflect that wind farms can impact freezing sea spray conditions: "Thus, it is crucial to understand the icing hazard across the mid-Atlantic."

Near-surface ice accretion onto vessels and areas where personnel work can induce life-threatening conditions and thus we believe it is crucial to study. This study was motivated by a specific question from wind industry colleagues who were interested and concerned about this.

82 What is "post-production effect" in this context?
We now refer to the "post-construction" of turbines to clarify that these model simulations include wind turbines: "Herein, we employ numerical weather prediction modeling to quantify the baseline offshore icing risk and the wind plant post-construction effects."

Methods:
157-161 Spray freezing and riming are two distinct phenomena and the paragraph is not sufficiently clearly introducing them as such.
Thank you for raising this issue. Accordingly, we have modified our method to account for many types of icing at the hub height instead of just precipitation as in

our original method. We have modified this paragraph and our methods to provide a clearer introduction of the process. Namely, at the hub height, we check for rain, snow, ice and a relative humidity greater than or equal to 100%:

"Due to the height constraint of sea spray particles, we consider both precipitation-based and in-cloud icing at the 138 m hub height by assessing different criteria for 1) the nonzero presence of liquid rain water (WRF variable "QRAIN") that may become supercooled at temperatures less than 0°C, 2) ice (WRF variable "QICE"), and 3) the aggregation from snow (WRF variable "QSNOW") (ISO, 2017; Parent and Ilinca, 2011). Further, we detect cloud or fog formation when 4) the relative humidity (RH) is greater than or equal to 100% following:

$$e_s = e_0 \exp\left[\frac{b(T - T_1)}{(T - T_2)}\right] \tag{1}$$

$$w_s = \frac{\epsilon e_s}{p - e_s} \tag{2}$$

$$RH = \frac{w}{w_s} \times 100\% \tag{3}$$

where $e_s$ is the saturation mixing ratio, $e_0$ is 6.112 mb, $b$ is 17.67, $T_1$ is 273.15 K, $T_2$ is 29.65 K, $T$ is the air temperature, $\epsilon$ is 0.622, $p$ is the atmospheric pressure, and $w$ is the mixing ratio (WRF output "QVAPOR") (Stull B., 1988). None of the aforementioned criteria must occur at the same time in order for FSS to occur. However, we require that one must occur in conjunction with an air temperature less than 0° C for a FSS event."

188 Do the convective rolls have any meaning in the context of sea spray freezing, or are they just relevant for the in-cloud icing? Please clarify.
Convective rolls can be used to identify cold air outbreaks and may also contribute to in-cloud icing. We have added a clarifying sentence: "Convective rolls can be used to identify cold air outbreak (CAO) (Atkinson and Wu Zhang, 1996; Geerts et al., 2022) and may also contribute moisture for in-cloud icing if the lifting condensation level is at or below rotor-swept heights."

194 Is perhaps the temperature at 2 meters meant here, and not at 10 meters? There are more occurrences of the 10 m temperature. Please check.
Thank you for asking about this. We have modified our methodology to assess the 2-m air temperature for CAO instead of the 10-m temperature.

Results:
221-226 Is this current dynamics supposed to help explaining the results regarding the freezing events. If yes, then proof is required, otherwise it is just speculation. If no, then it is not necessary.

Current dynamics feed cold water from the Western Maine Current into the mid-Atlantic bight. This source of cold water promotes colder SSTs (Beardsley et al., 1985; Bigelow, 1915; Chapman et al., 1986; Linder and Gawarkiewicz, 1998). We have added that the faster wind speeds in this region (as assessed in the NOW-23 dataset (Bodini et al. 2023) also increase the icing hours:

"The Long Island Sound is flanked by land to the north and south which amplifies the presence of cold air. In addition, mean wind speeds maximize to the east of Cape Cod and Nantucket (Bodini et al., 2023) which increase the number of hours that wind-generated spray is present. Finally, the cyclonic current in the Gulf of Maine transports water southward. East of Cape Cod, this current bifurcates around the Georges Bank, and a branch feeds cold water into the mid-Atlantic (Chapman et al., 1986). The number of icing hours may be further exacerbated when predominant northerly winter winds instigate onshore Ekman transport toward the coast, which is favorable for downwelling (Shcherbina and Gawarkiewicz, 2008b). However, downwelling is not always supported, as the mixed layer stratification is dominated by salinity (Shcherbina and Gawarkiewicz, 2008a), leaving a cold pool near the surface."

229-230, Figure 2 caption. Please normalize the color scales on the two plots so that they can be compared. For example, show the number of hours per season.
We have normalized the color bars between the figures for comparison. We now show the mean number of hours per season in panel (b) as requested.

[Figure]

*Figure 2. The number of hours FSS conditions occur at 10 m during (a) the November 2019 to March 2020 period in NWF and (b) the mean November to March period from 2000 to 2020 in NOW-23. Lighter contouring indicates more freezing hours. Red dots represent turbine locations but do not exist in (a) or (b) and are shown for reference.*

234 "freezing conditions" is slightly vague, especially since you calculate the magnitude as well. Could you perhaps use the magnitude even more?

This figure is not assessing PR, but instead the number of icing hours based on our criteria to detect FSS, and as such there is no magnitude. We have replaced "freezing" with "icing" for clarification. The next section discusses the severity of icing events using the PR equation.

236 Again, more than the area (12 times the wind plants), the severity of the freezing events would be more important to discuss here.
Here we discuss the spatiotemporal variability in the number of hours that icing events occur based on the three criteria (wind speed, air temperature, SST) which do not have an associated severity metric.  We later assess the icing events using PR, and discuss severity based on ice accretion there.

245 Figure 3. It is not immediately clear where the zoom fits. Please consider redrawing.
We have redrawn the figure to make it clearer where the zoom fits and have also added a more explanative caption:

[Figure]

Figure 3. The maximum number of FSS hours over the OCS (a) annually and (b) seasonally in NOW-23. The zoomed orange cutout shows the seasonal variation over the 2019–2020 winter.

260 "253 hours" seems inconsistent with the total which is  182 (or 187). Please check or clarify.

We have clarified that 182 hours comes from our FSS criteria in the topic sentence: "The 2019–2020 winter season was one of the mildest compared to other winters (Figure 3a), as assessed using the FSS detection criteria (Section 2.3)."

In Section 3.2, we already mention that 253 hours come from PR.

261 "light ice" here helps to slightly resolve my comment about the freezing severity, above.
We have also added a discussion on why the 2003-2004 winter season had the most severe winter PR based on teleconnection patterns.

265 It would be more appropriate to express the pressure gradient in hPa per 100 km (the value 4 would then mean 4 times the geostrophic wind speed of 10 m/s) - just a suggestion.
Thank you for the suggestion. We agree that this makes for easier interpretation. We have modified the sentence to read: "The largest pressure gradient forces occurred during the two January events reaching 4 hPa per 100 km, or roughly 4 times the pressure gradient force required for a 10 m s−1 geostrophic wind in the midlatitudes."

266 The reference is weird. The geostrophic wind and how it is calculated was first mentioned in 1857. In meteorology, work of e.g. Bjerknes would also be a meaningful reference. Again, the references should point to the earliest appropriate publication, not to a random one.
We have added description of the dynamics and have pointed to an original paper: "In the Northern Hemisphere, winds flow with higher pressure to the right and lower pressure to the left (Wallace and Hobbs, 2006). This flow regime results from the balance between the pressure gradient force and the Coriolis force, which is a force introduced into the equations of motion to account for acceleration on a non-inertial rotating reference frame (Ferrel, 1856)."

281 The meaning of CAO in the context of freezing is not clear. Does it matter if an event is called CAO? Especially since one uses the same variables to calculate if an event is CAO, and if there is freezing.
This misunderstanding has likely resulted from our use of "freezing" and "icing" in the text.  Yes, the same variables are used in determining CAO and freezing conditions. However, there are additional variables used to determine *icing* conditions and we have carefully edited our use of "freezing" and "icing" in the text.
Here we show that for this winter period, the introduction of cold air during CAO events was the primary driver for icing conditions. The reason it matters if an

event is called CAO is that CAO events can be forecast multiple days in advance which can increase the forecast lead time for freezing sea spray conditions.

287-288 True statement, the grid efficiency does suffer from high temperature, but is irrelevant in the context of freezing. Please consider removing, or explain why this is important for this paper.
In this sentence we are referring to the heating of homes and businesses, not outdoor air temperatures and their effect on the transmission.  We have clarified this point: "This wind speed–temperature dynamic can pose a challenge for grid planners if wind energy generation reduces during periods of high demand for residential and commercial heating, especially in a future scenario with electrification of space heating."

309 The total effect of up to -0.041 K is so small that it should perhaps be pointed out
even more, how small the effect of wind farms to the freezing is.
We have modified the topic sentence of this paragraph to better portray this result: "The near-surface cooling effect by rotor turbulence provides a subtle effect on freezing conditions".

315 Figure 5. Which height above the surface is this?
This figure shows a latitude by height cross section. Heights are shown across the y axis up to 300 m.

320 Icing or freezing, blades or ship? Please clarify.
Icing is correct here because slower wind speeds reduce both convective heat transfer and wind-generated spray. In unstable conditions, wakes expand and extend down to the surface so both blades and the ship.  This sentence has been restructured as follows: "The reduction of wind speeds in the wake modifies the chance for icing within the rotor-swept area and near the surface by reducing the production of white-capped waves and the wind-induced tearing of spray off waves."

322 Here you say freezing. It is really not OK to use icing and freezing interchangeably like this!
We have edited the text to correctly refer to freezing in regard to the temperature and to icing in regard to conditions with freezing temperatures and moisture availability.

337 13 hours, compared to what? Please express as fraction.
All WFP changes are relative to NWF. We have clarified this point in the topic sentence of the paragraph: "Despite near-surface cooling, net FSS conditions in WFP

occur less often than in NWF when diagnosed using wind speed, air temperature, and SST criteria because of the wake wind speed reduction."

Thank you for the suggestion to express this as a fraction, but we initially reported the changes to icing hours as fractions and percentages. However, scientific colleagues at conferences and industry collaborators have repeatedly requested us to express our findings as the number of hours to be more intuitive.

345-346 "… flow acceleration is present …". "may be present" would be more accurate, it is not relevant for his paper, so why mention it.
We agree that this point may not be relevant and, in addition with your next comment, have removed this sentence.

347-348 The statement about the numerical noise seems to negate the rest of the analysis. The physics of freezing is correctly captured in the models, and the results are consistent. It is true that WFP can introduce noise, even at the opposite side of the planet in e.g. MPAS model. Please provide more results supporting the numerical noise hypothesis, or consider removing the statement.
We have incorporated hourly averaging to reduce the effects of noise following our original approach with this simulation output (Rosencrans et al., 2024). The numerical noise is almost impossible to see, so we have removed this statement.

[revised manuscript text omitted]

---

## Author Comment (AC2)

Reviewer comments appear in **black** and author responses in **blue**.

**Reviewer 2**

Dear authors,

The article is well written and interesting to read. The subject is original within the field of wind energy research and is of relevance for the future offshore wind farm development of the east coast in the USA. The results show that there is considerable icing risk in the mid-Atlantic offshore wind farm areas and that the effect of wind farm wakes on icing risk is minimal. The manuscript would benefit by considering the points below.

We thank the reviewer for devoting time to our article and for providing suggestions that improve the work.

Specific comments:

Regarding the two sentences and corresponding references:

"Some observations indicate that excessive icing can reduce torque enough that blade rotation stops entirely, causing up to 80 % reduced power production for a single turbine".

"Some turbines have icing detection and mitigation technology included at added cost, although current strategies need improvement (Madi et al., 2019)."

I believe these statements are a bit outdated, especially seen in the light that the paper is mainly focusing on future wind energy scenarios. I do not want to make an advertisement for specific solutions, but there are many de-icing, or anti-icing solutions where power reduction can be avoided (see e.g. https://www.iqpc.com/media/1001147/37957.pdf, https://wicetec.com, https://www.video.vestas.com/video/21313125/vestas-anti-icing-system). I suggest that the text should be updated to state that it will be important to include some proper anti-icing solutions.

Thank you for the suggestion. Per reviewer 1's feedback, we have changed the first sentence to "One study found that excessive icing induced a power loss of 63 % for a single turbine over a 51-h icing event (Gao and Hu, 2021)".

We believe it is still necessary to provide context for the issue, that icing can reduce power production. Our Madi et al. (2019) reference lists a variety of different icing mitigation strategies and we have added IEA Task 19 as well. Given that we focus on a future scenario we have modified the sentence to not focus on the setbacks of ice mitigation strategies, as these strategies are improving and will continue to improve in the future: "Despite the energy losses in some studies, various strategies can mitigate or even prevent ice accretion altogether (Madi et al., 2019)."

It becomes clear after reading section 2.1 that the NOW-23 data set is also based on WRF runs. I would suggest mentioning that at the beginning of section 2.1.
We now mention that this modeling data set is based on WRF runs in sentence two of the paragraph, before describing the model setup: "This data set quantifies wind resources spanning all offshore regions of the United States for more than 20 years using the Weather Research and Forecasting (WRF) model version 4.2.1 (Powers et al., 2017)."

Table 1 is a bit strange. Maybe there should be a column with simulation type number 1-3, or something similar that could be referred to in the text. "Turbine type" should be "turbine rated power", or if you want to keep turbine type then mention the type. The period does not need a column (as it is the same for all simulations) could be mentioned in the figure caption.
Thank you for the suggestions, which improve the readability and utility of the table. We have incorporated several of your suggestions:
- The column title "Turbine type" has been changed to "Turbine rated power".
- We now mention the period of analysis in the caption.
- Because we do not use the TKE 0 % simulations in this study, we have removed their mention from the table.
- We added a column titled "Acronym" for clarity. Further, we explicitly mention that the acronym "WFP" will refer to the simulation with 100 % added TKE: "Thus, for the remainder of this article we refer to the 100 % added TKE simulation as "WFP".

*Table 1. List of WRF simulations characterized by turbine characteristics. The simulation period spans 01 September 2019 to 01 September 2020.*

| Simulation type | Acronym | Turbine rated power | Added TKE | # Turbines |
|---|---|---|---|---|
| No Wind Farms | NWF | N/A | N/A | 0 |
| Wind Farm Parameterization | WFP | 12 MW | 100 % | 1,418 |

Line 152: Why can SST be replaced by skin temperature?
The skin temperature in WRF is the temperature of the surface, whether the surface is land or ocean. So, the skin temperature is the same as the SST and we mask the skin temperature to only retrieve data over the ocean. We do so because the SST field is very coarse.

Have the authors investigated how the results would change if you would use the most conservative thresholds? It would be beneficial to include a sensitivity study about that, e.g. on a small subset of data or at the POI.

Thank you for this excellent suggestion. We have added a sensitivity analysis to Appendix section B, testing between air temperatures of -1.7°C and -2°C and between sea surface temperatures of 5°C and 8.9°C. Our findings are that the maximum number of icing hours do not change much, but the regional variability changes considerably depending on the thresholds used:

"As discussed in Section 2.3, we detect FSS conditions using common thresholds for the meteorological conditions (Dehghani-Sanij et al., 2017; Guest and Luke, 2005; Line et al., 2022). These criteria require strong wind speeds greater than 9 m s$^{-1}$, cold air temperatures below −1.7° C, and cold SSTs less than 7° C. As reviewed by Dehghani-Sanij et al., (2017), FSS conditions are promising when the air temperature is below either −1.7° C or −2° C to account for the lower freezing point of saline ocean water; the salt content of which determines this threshold. Although SST thresholds of 5° C or 7° C are prevalent, a threshold up to 8.9° C has been used (U.S. Navy, 1988). As such, we quantify some of the uncertainty by calculating the number of hours that FSS conditions occur using conservative thresholds, which produce fewer icing hours (FEWER), and liberal thresholds, which promote more icing hours (MORE) (Table B1). As there is wider agreement regarding the wind speed threshold (Dehghani-Sanij et al., 2017; Guest and Luke, 2005; Line et al., 2022; Monahan et al., 1983; Monahan and MacNiocaill, 1986; Ross and Cardone, 1974), we hold it constant. Due to computational constraints, we only assess the number of icing hours throughout the domain at 10 m and during January 2020 because it has the greatest number of icing hours.

*Table B1. Icing detection criteria by sensitivity analysis type.*

| Acronym | Air temperature | Sea surface temperature | Wind speed |
|---|---|---|---|
| **FEWER** | <−2° C | <5° C | >9 m s$^{-1}$ |
| **MORE** | <−1.7° C | <8.9° C | >9 m s$^{-1}$ |

As expected, more conservative thresholds produce fewer FSS hours and vice versa (Fig. B1a,b,c). In FEWER, the meteorological conditions conducive to icing maximize at 60 hours. Using more liberal criteria in MORE, the maximum number of hours increases to 67. Despite the small change in the maximum number of hours FSS occurs, the regional variation is large; the area covered by icing conditions increases from 8,924 km2 to 135,244 km2 from FEWER to MORE, or roughly 15 times greater than FEWER, or 2.2 times greater than our production set of criteria. Regional variability follows SST patterns and only occurs in FEWER where the SST is relatively cold in the Long Island Sound and Nantucket Sound (Table B1b), as discussed previously.

[Figure]

*Fig. B1. The number of hours FSS conditions occur during January 2020 at 10 m in NWF using thresholds for (a) FEWER, (b) MORE, and (c) the (FEWER-MORE) difference. Lighter contouring indicates more freezing hours in (a) and (b). Darker blues represent a larger reduction in number of hours in (c). Turbine locations are shown as red dots in (a) and (b) and as black dots in (c).*

Why is the acronym for predictability chosen as PPR?
Thank you for this thoughtful question. This icing predictability equation has been reported in the literature as PPR (Dehghani-Sanij et al., 2017; Guest and Luke, 2005). However, this may be a typo, as the original authors used "PR" (Overland, 1990; Overland et al., 1986). To squash this recurrence, we have changed the acronym to "PR" throughout the article.

Fig. 2: It's a little bit confusing that the turbine locations are shown for these simulations that were performed without turbines. Could you state in the caption that the locations are shown for illustrative purposes, but were not included in the simulation results?
We have clarified this point in the figure caption: "Red dots represent turbine locations but do not exist in (a) or (b) and are shown for reference."

Fig. 7: Is it percentage or difference in whole hours? If percentage, the color bar label needs to state that by adding e.g. [%]. Could you include smaller intervals on the color scale, so it's possible to see more variation?
We now show the change in number of hours for consistency with other figures and have changed the figure caption as follows: "The (WFP-NWF) change in number of FSS hours at 10 m November 2019 to March 2020. Blue contours indicate a reduction."

We have also doubled the contour interval to enhance the granularity:

[Figure]

**References**

Dehghani-Sanij, A. R., Dehghani, S. R., Naterer, G. F., and Muzychka, Y. S.: Sea spray icing phenomena on marine vessels and offshore structures: Review and formulation, Ocean Engineering, 132, 25–39, https://doi.org/10.1016/j.oceaneng.2017.01.016, 2017.

Gao, L. and Hu, H.: Wind turbine icing characteristics and icing-induced power losses to utility-scale wind turbines, Proceedings of the National Academy of Sciences, 118, e2111461118, https://doi.org/10.1073/pnas.2111461118, 2021.

Guest, P. and Luke, R.: The Power of Wind and Water, Mariners Weather Log, https://www.vos.noaa.gov/MWL/dec_05/ves.shtml, 2005.

Line, W. E., Grasso, L., Hillger, D., Dierking, C., Jacobs, A., and Shea, S.: Using NOAA Satellite Imagery to Detect and Track Hazardous Sea Spray in the High Latitudes, Weather and Forecasting, 37, 351–369, https://doi.org/10.1175/WAF-D-21-0137.1, 2022.

Madi, E., Pope, K., Huang, W., and Iqbal, T.: A review of integrating ice detection and mitigation for wind turbine blades, Renewable and Sustainable Energy Reviews, 103, 269–281, https://doi.org/10.1016/j.rser.2018.12.019, 2019.

Monahan, E. C. and MacNiocaill, G.: Oceanic Whitecaps And Their Role in Air-Sea Exchange Processes, D Reidel Publishing Company, e-ISBN-13: 978-94-009-4668-2, https://link.springer.com/book/10.1007/978-94-009-4668-2, 1986.

Monahan, E. C., Fairall, C. W., Davidson, K. L., and Boyle, P. J.: Observed inter-relations between 10m winds, ocean whitecaps and marine aerosols, Quarterly Journal of the Royal Meteorological Society, 109, 379–392, https://doi.org/10.1002/qj.49710946010, 1983.

Overland, J. E.: Prediction of Vessel Icing for Near-Freezing Sea Temperatures, Weather and Forecasting, 5, 62–77, https://doi.org/10.1175/1520-0434(1990)005<0062:POVIFN>2.0.CO;2, 1990.

Overland, J. E., Pease, C. H., Preisendorfer, R. W., and Comiskey, A. L.: Prediction of Vessel Icing, Journal of Applied Meteorology and Climatology, 25, 1793–1806, https://doi.org/10.1175/1520-0450(1986)025<1793:POVI>2.0.CO;2, 1986.

Powers, J. G., Klemp, J. B., Skamarock, W. C., Davis, C. A., Dudhia, J., Gill, D. O., Coen, J. L., Gochis, D. J., Ahmadov, R., Peckham, S. E., Grell, G. A., Michalakes, J., Trahan, S., Benjamin, S. G., Alexander, C. R., Dimego, G. J., Wang, W., Schwartz, C. S., Romine, G. S., Liu, Z., Snyder, C., Chen, F., Barlage, M. J., Yu, W., and Duda, M. G.: The Weather Research and Forecasting Model: Overview, System Efforts, and Future Directions, Bulletin of the American Meteorological Society, 98, 1717–1737, https://doi.org/10.1175/BAMS-D-15-00308.1, 2017.

Ross, D. B. and Cardone, V.: Observations of oceanic whitecaps and their relation to remote measurements of surface wind Speed, Journal of Geophysical Research (1896-1977), 79, 444–452, https://doi.org/10.1029/JC079i003p00444, 1974.

U.S. Navy: U. S. Navy Cold Weather Handbook for Surface Ships, Surface Ship Survivability Office, https://media.defense.gov/2021/Feb/25/2002588484/-1/-1/0/CG%20070%20-%20US%20NAVY%20COLD%20WEATHER%20HANDBOOK.PDF, 1988.

---

## Author Response (AR2)

Reviewer comments appear in **black** and author responses in **blue**.

Dear authors,
You have addressed the reviewer comment very well and the article has improved by the review process.
I only have two comments that I would like to be addressed/fixed before publication:

1) FSS in used in the abstract without explanation. It is first spelled out in the introduction. Either use 'freezing sea spray' in the abstract or introduce the acronym there. It's generally preferable to avoid acronyms in the abstract for clarity. Additionally, please check that all acronyms are correctly introduced.

Thank you for pointing this out. We have clarified on line 19 that the data set is used to assess freezing sea spray events and have replaced the acronym "FSS" with "freezing sea spray" on line 24.

2) Regarding the thresholds used to define FSS conditions: have these thresholds been validated? If so, it would be helpful to reference that validation in the text. If not, could you briefly discuss how the lack of validation might impact your results?

We have provided a brief discussion of how our lack of validation impacts the results in Appendix B, to further explain motivation for the sensitivity analysis, on new lines 527-532:

"Although these thresholds were derived from observations aboard ships, the observations are sparse and have not been validated in the mid-Atlantic. Using higher air and sea surface temperature thresholds may cause an overestimation of the number of freezing hours when mid-Atlantic waters are more saline, for example, during periods with higher evaporation rates. Further, large water droplets have a higher chance of becoming runoff instead of freezing. Thus, our results may overestimate the number of icing hours where significant wave breaking and bubble bursting occur and underestimate the number of icing hours in calmer waters."